# AlphaDesign: a de novo protein design framework based on AlphaFold

Michael A Jendrusch [1,2], Alessio L J Yang [1,2], Elisabetta Cacace[1,3], Jacob Bobonis [4], Carlos G P Voogdt [1], Sarah Kaspar [1], Kristian Schweimer [5], Cecilia Perez-Borrajero [1], Karine Lapouge [6], Jacob Scheurich[6], Kim Remans [6], Janosch Hennig [1,5], Athanasios Typas [1], Jan O Korbel [1]✉ & S Kashif Sadiq [1,7]✉

## Abstract

**De novo protein design is of fundamental interest to synthetic biology, with a plethora of computational methods of various degrees of generality developed in recent years. Here, we introduce AlphaDesign, a hallucination-based computational framework for de novo protein design developed with maximum generality and usability in mind, which combines AlphaFold with autoregressive diffusion models to enable rapid generation and computational validation of proteins with controllable interactions, conformations and oligomeric state without the requirement for class-dependent model re-training or fine-tuning. We apply our framework to design and systematically validate in vivo active inhibitors of a family of bacterial phage defense systems with toxic effectors called retrons, paving the way towards efficient, rational design of novel proteins as biologics.**

**Keywords** AlphaFold; Computational Biology; Machine Learning; De Novo Protein Design; Anti-phage Defense Proteins
**Subject Categories** Biotechnology & Synthetic Biology; Computational Biology; Structural Biology

## Introduction

Evolution has only explored an infinitesimal portion of the potential protein sequence landscape (Huang et al, 2016). Enormous potential, therefore, exists in unlocking the principles of protein folding (Lindorff-Larsen et al, 2011) and accurate structure prediction to design and engineer novel proteins that can exploit this vast space. Established protein engineering methods have largely focused on tuning naturally occurring proteins through iterative experimental selection processes such as directed evolution

(Dougherty and Arnold, 2009). Recently there has been rapid progress in computational methods for protein design. Many approaches to design protein backbones, such as hallucination-based methods (Norn et al, 2021; Frank et al, 2024; Tischer et al, 2020; Wang et al, 2022; Wicky et al, 2022) and denoising diffusion probabilistic models (DDPM) (Watson et al, 2023; Lisanza et al, 2024; Ingraham et al, 2022; Yim et al, 2023), have been developed, with diffusion models, so far, outperforming most hallucination-based approaches in design quality. Combined with neural networks capable of generating native-like sequences folding into any given protein backbone structure (Ingraham et al, 2019; Dauparas et al, 2022) these methods have enabled rapid design of proteins with a variety of properties (Wicky et al, 2022; Yeh et al, 2023). Most current studies on machine-learning-based protein design methods focus mostly on de novo protein design tasks, such as de novo monomer design (Norn et al, 2021; Frank et al, 2024; Yim et al, 2023; Lin and AlQuraishi, 2023; Lisanza et al, 2024) or homooligomer design (Wicky et al, 2022). This functionality has been extended to site-specific protein binder design by more recent DDPM-based methods (Watson et al, 2023) but these require considerable engineering and training efforts to arrive at a usable model.

While these methods and design capabilities have great potential, they currently address only a fraction of the functional properties of natural proteins. Many proteins exhibit multiple functionally relevant states, either by binding multiple partners and/or through exhibiting conformational changes across the course of their natural function.

While prior work has investigated the potential of machine learning for designing sequences compatible with multiple conformations, such work has largely relied on the use of manually-designed structural blueprints (Praetorius et al, 2023; Guo et al, 2024), instead of generating multi-state protein backbones and sequences fully de novo.

Here we describe AlphaDesign, a protein design framework, the versatility of which enables both standard (single-state) and multi-state protein design by combining hallucination and diffusion-

---

[1]European Molecular Biology Laboratory (EMBL), Heidelberg, Germany. [2]Collaboration for joint PhD degree between EMBL and Heidelberg University, Faculty of Biosciences, Heidelberg, Germany. [3]Institute of Microbiology, Department of Biology, ETH Zurich, Zurich, Switzerland. [4]Division of Microbial Ecology, Centre for Microbiology and Environmental Systems Science, University of Vienna, Vienna, Austria. [5]Chair of Biochemistry IV, Biophysical Chemistry, University of Bayreuth, 95447 Bayreuth, Germany. [6]Protein Expression and Purification Core Facility, EMBL, Heidelberg, Germany. [7]DenovAI Biotech Ltd., Rehovot 7670104, Israel. ✉E-mail: jan.korbel@embl.de; kashif@denovai.com

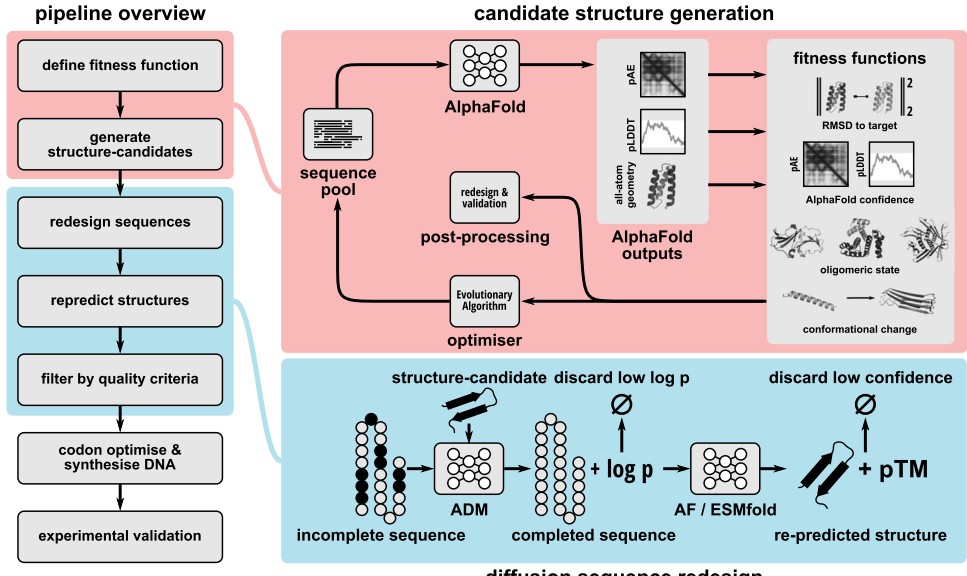

**Figure 1. Overview of the AlphaDesign pipeline.**

The AlphaDesign pipeline consists of two main stages: candidate structure generation (red) and sequence redesign (blue). In the first stage, a pool of random amino acid sequences has their structures and confidence measures (pLDDT, pAE) predicted using AlphaFold. These are combined into a fitness function, which in turn is maximised using an evolutionary algorithm. Once a sequence's fitness reaches a threshold, its predicted structure (raw structure) is returned for sequence redesign and computational validation. The second stage uses an autoregressive diffusion model (ADM) trained on the PDB to generate new sequences for each structure from the first stage. These are then re-predicted using multiple structure predictors (predicted structure). Sequences with low-confidence predicted structures or high RMSD to the raw structure are discarded. Source data are available online for this figure.

based approaches. Our approach leverages AlphaFold (AF) (Jumper et al, 2021) for structure prediction and autoregressive diffusion models (ADM) (Hoogeboom et al, 2022) for sequence optimisation. To develop a method capable of designing proteins with any given set of properties, we combine AF structure predictions and confidence measures into fitness functions encoding various protein design tasks. Given a sufficiently powerful optimiser, the determining factor for protein design success lies entirely in the fitness function. While prior work on hallucination-based protein design optimised various confidence measures for specific protein design tasks, we expand on this line of research by developing modular fitness functions which can be combined to achieve protein complex design, binder design, as well as multi-state protein design.

We utilise an evolutionary algorithm (Sinai et al, 2020) as an optimiser to search for amino acid sequences maximising these fitness functions (Fig. 1 (red)). This ensures that we can, in principle, optimise proteins for any combination of properties that can be predicted by AF, including changes in conformation between monomeric and complex structures as well as binding to multiple target proteins.

Optimising sequences for high structure prediction confidence runs the risk of generating adversarial examples for AF which are unlikely to be recombinantly expressed in *E. coli* or adopt the designed tertiary structure. Indeed, prior work showed that amino acid sequences designed in a similar way are unlikely to be expressed (Wicky et al, 2022; Goverde et al, 2022; Verkuil et al, 2022).

In order to avoid this issue and to overcome major challenges in the field associated with solubility and expressibility of de novo

designed proteins (Wicky et al, 2022; Frank et al, 2024), we further redesign the sequences of high-fitness candidate proteins using an ADM (Hoogeboom et al, 2022), trained on the PDB (Fig. 1 (blue)). This ADM generates a set of new native-like sequences for each structure returned by the evolutionary algorithm. We find that sequence redesign is a crucial ingredient for successful protein design with AlphaFold. We combine this design pipeline with automated computational validation and design selection by predicting the structures of our designs with AlphaFold and ESMfold (Wu et al, 2022). As de novo designed proteins do not have evolutionary information attached to them, we predict their structures from their sequence alone without providing a multiple sequence alignment (MSA). While AlphaFold is known to lose accuracy on natural proteins in the absence of MSA information, prior work has shown that single-sequence prediction is effective in evaluating design success for de novo designed proteins and has become the de facto standard for computational evaluation of protein designs (Bennett et al, 2022; Wicky et al, 2022; Martin et al, 2024).

In contrast to most previous approaches, our hybrid framework produces results that are competitive with DDPM-based methods, whilst retaining the full flexibility of hallucination-based approaches. Only recently Frank et al demonstrated a comparable approach yielding similar results to RFDiffusion on de novo protein design (Frank et al, 2024). We show state-of-the-art performance of AlphaDesign on computational de novo protein design benchmarks and demonstrate hallucination-based protein binder design beyond short peptides (Bryant and Elofsson, 2022; Rettie et al, 2023), which has been shown to be possible only in one concurrent work on protein design (Pacesa et al, 2024). Our method is versatile enough

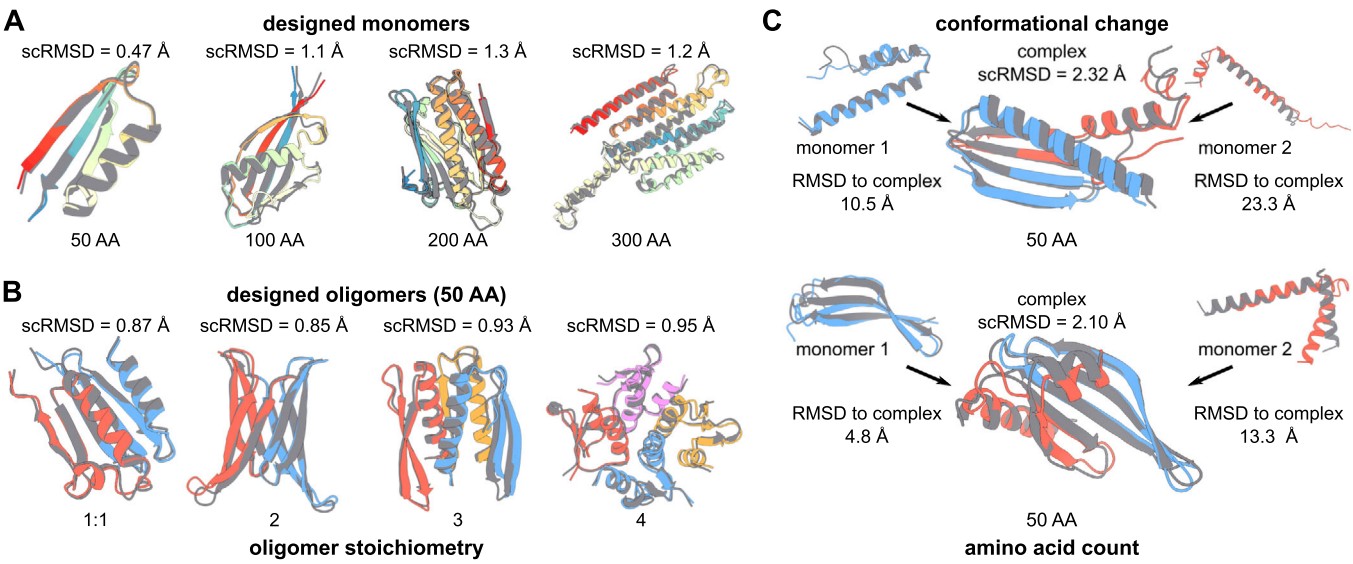

**Figure 2. De novo designed proteins.**

(A, B) example structures of de novo designed monomers with sequence length from 50 to 100 amino acids and complexes of 2 to 4 monomers with a sequence length of 50 amino acids. The raw structures for each design are shown in grey; predicted structures are shown in colour. scRMSD between raw and predicted structures are reported for each design. (C) structures of designed monomers which are predicted to change conformation upon complex formation. The raw structures for each design are shown in grey; predicted structures are shown in colour. scRMSD and RMSD between monomeric and complex states are reported for each pair of designs. Source data are available online for this figure.

to design proteins with various properties for example, proteins that are predicted to change conformation upon binding, as well as proteins predicted to bind multiple homologues of a protein, all without the requirement for model re-training or fine-tuning. These features are absent from other machine learning-based tools for protein design. In addition to extensive computational validation using accurate structure prediction tools and all-atom molecular dynamics simulations, we demonstrate the real-world applicability of our method by systematically designing and experimentally validating inhibitors of the toxic effector RcaT found in the most prevalent sub-family of retrons, a newly discovered bacterial phage defense system (Millman et al, 2020; Gao et al, 2020; Bobonis et al, 2022). We confirmed in vivo activity for 17 out of 88 designs, as well as expression and fold using NMR structure determination for 2 designs.

## Results

### Generation of designable monomers and oligomers

As a first step to demonstrate the capabilities of our framework, we generated raw sequences and structures of globular monomeric proteins, oligomers and proteins predicted to change conformation upon binding (Fig. 2A–C). We generated monomers by maximising AlphaFold confidence, subject to a radius penalty (Appendix Table S2, $L_{denovo}$). We then used an ADM to produce a set of redesigned sequences for each generated raw structure.

We computationally assessed design success by predicting the structures of these sequences with AF and ESMfold (Evolutionary Scale Modelling) (Wu et al, 2022). We used ESMfold as an

additional protein structure predictor for validation in order to ensure that our designed sequences were not biased for AlphaFold. A designed sequence was deemed successful if the predicted local distance difference test (pLDDT) was >70 and the root mean squared deviation (self-consistent RMSD; scRMSD) between the raw generated and predicted structure was <2.0 Å—thresholds chosen in agreement with the state of the art of computational protein design approaches (Watson et al, 2023). Structure prediction showed very good agreement between generated and predicted structures with RMSD < 2 Å for a majority of designs (Figs. 3A and EV1A,B). More precisely, 97.6% and 98.6% of designed 50 AA monomers were successful according to AF and ESMfold, respectively (100 AA: 92.8% and 98.6%, 200 AA: 85.3% and 89.3%, 300 AA: 72.4% and 86.2%).

Similarly, we were able to design heterodimers and homo-oligomers by maximising AlphaFold confidence for jointly predicting the structure of multiple amino acid sequences. We once again obtained high rates of success for designed heterodimers and homooligomers (Figs. 2B and 3A) with 79.5% of designed 50 AA heterodimers (respectively, homodimers: 72.4%, trimers: 74.3%, tetramers: 70.1%) deemed successful using AF, compared to 60.0% using ESMfold (respectively, homodimers: 50.8%, trimers: 34.2%, tetramers: 47.8%). This difference in success-rates mirrors previous results, and suggests that ESMfold performs worse at complex structure prediction (Wu et al, 2022) and predicts designed structures even for suboptimal de novo designed sequences (Martin et al, 2024).

To test the limits of our fitness function-based design approach, we optimised pairs of amino acid sequences to minimise the template modelling score (TM score) between each sequence predicted as a monomer, and the pair of sequences predicted as a

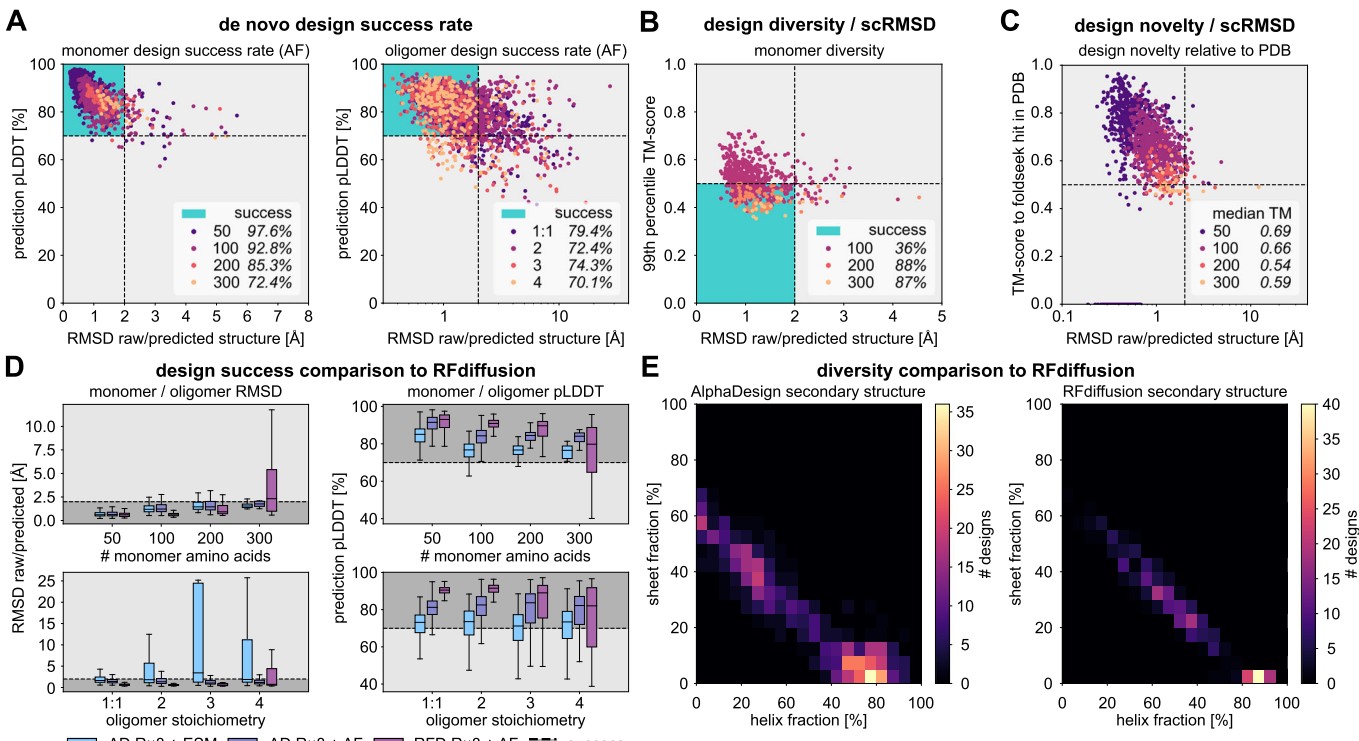

**Figure 3. Design success and diversity.**

(**A**) Scatter plots of scRMSD and pLDDT for monomers (left) and oligomers (right). The threshold for design success is marked with dashed lines for both pLDDT and scRMSD. The region where designs are deemed successful in terms of both scRMSD and pLDDT is marked in blue. Success rates are reported for each size of monomer and each complex stoichiometry. AlphaDesign sample sizes throughout this figure for monomers, 50 AA: $N = 1037$; 100 AA: $N = 540$; 200 AA: $N = 75$; 300 AA: $N = 29$; oligomers 1:1: $N = 1173$; 2: $N = 1267$; 3: $N = 319$; 4: $N = 251$. (**B**) Scatter plot of redesigned sequence scRMSD and TM-scores computed using TMalign to the closest foldseek hit in the PDB. Designs are coloured by number of amino acids and median TM-scores are reported for each sequence length. (**C**) Scatter plot of redesigned sequence scRMSD and 99th percentile TM-score to any other design of the same size. TM-Score < 0.5 indicates that the two compared proteins adopt a distinct fold. Designs count as successful (turquoise region) in terms of diversity if TM-Score < 0.5 and scRMSD < 2.0 Å or pLDDT > 70.0. Percentages of successful designs are reported. (**D**) Measures of success for designed monomers. Comparison of design self-consistent RMSD and pLDDT for best-of-8 redesigned AlphaDesign sequences (AD R×8 + ESM/AF) and RFDiffusion + ProteinMPNN designed sequences (RFD R×8 + AF). Values for monomers are shown in the top row and values for oligomers in the bottom row. The centre line of each box corresponds to the median, box edges to the upper and lower quartiles, and whiskers to the lowest and highest data points excluding outliers (within 1.5 times the interquartile range). $N = 100$ samples were drawn from RFdiffusion for each benchmark. (**E**) Binned secondary structure distributions for de novo designed monomers of sizes 100–300 amino acids using AlphaDesign (left) and RFDiffusion (right). The fraction of alpha-helix amino acids is marked on the x-axis; the fraction of beta-sheet amino acids on the y-axis. Source data are available online for this figure.

dimer (Fig. 2C). TM-score measures the structural similarity between two protein structures, with scores below 0.5 indicating different folds (Zhang and Skolnick, 2004). Thus, a pair of sequences with low TM-score between monomeric and complex state is essentially predicted to change conformation upon binding. While our approach was able to generate designs predicted to change conformation, it yielded lower success-rates compared to unconstrained de novo design (Fig. EV2A,B).

Lower TM-scores between the monomeric and bound state of a design correspond to a more pronounced change in conformation. By penalising designs with TM-scores above a maximum threshold (TM-score < 0.1, 0.3, 0.5), we were able to control the degree of conformational change. For a maximum TM-score of 0.1, we were able to recover 3.6% of successful designs (respectively, TM-score < 0.3: 12.5%, TM-score < 0.5: 12.0%). This shows that our method can design proteins predicted to change conformation in a highly controllable manner, as assessed by state-of-the-art computational validation.

In addition to achieving a high de novo design success rate, our method also produces proteins with diverse and novel structures with folds ranging from all-$\alpha$-helical to all-$\beta$-sheet (Fig. 2A,B). Further analysis shows that designs are diverse relative to each other (Fig. 3B) and a high fraction are novel, relative to the PDB (Fig. 3C), with median similarity as measured by TM-score (Zhang and Skolnick, 2004) decreasing with sequence length. Taken together, these results indicate that our method is indeed capable of de novo designing diverse and novel proteins.

## De novo protein design is competitive with state-of-the-art approaches

In order to evaluate our approach relative to the state of the art, we compared the quality of our designs to RFDiffusion (Watson et al, 2023). We found our designed sequences and structures showed competitive performance across both monomers and oligomers of various sizes (Fig. 3D) as measured by scRMSD and pLDDT. In total,

the majority of ADM-redesigned sequences are successful with median scRMSD and pLDDT close to RFDiffusion, showing a slight improvement for 300 AA monomers and homotetramers (Fig. 3D). In addition, our approach generates structures with a greater variety in secondary structure content compared to RFDiffusion (Fig. 3E). This indicates that our method produces comparable or improved results to the state of the art protein diffusion model, while representing a hallucination-based approach.

## Computational complexity

We assessed the computational complexity of our method for de novo monomer design across different monomer sizes from 50 to 300 amino acids. We found that the number of design iterations increases rapidly with both the desired fitness of a monomer and its size (Appendix Fig. S1A). Fitting a polynomial to the data, revealed that the time needed per design on a single A40 GPU scaled as $O(N^{3.8})$ with $N$ the length of the amino acid sequence (Appendix Fig. S1B). Our method at least partly inherits this complexity from AlphaFold, which scales as $O(N^3)$ with protein sequence length. This suboptimal complexity is a shortcoming of our method, compared to diffusion-based approaches (Table 1). However, concurrent work has shown that this can be remedied by using different sequence optimisers (Frank et al, 2024) as well as more light-weight protein structure predictors (Jeliazkov et al, 2023).

## Sequence redesign is crucial for design quality

To assess the impact of ADM redesign on de novo design quality, we evaluated both raw and redesigned outputs of AlphaDesign in terms of pLDDT, scRMSD computed using AF and ESMfold, interface predicted aligned error (ipAE) (Evans et al, 2021) and predicted solubility (Sormanni et al, 2015). We observed an increase in predicted solubility (Sormanni et al, 2015) for 98.6% of redesigned monomers over raw sequences obtained by optimisation (Fig. 4A), coupled with a decrease in ESMfold scRMSD for 92.2% of designs (Fig. 4B).

Compared to ADM-redesigned sequences, scRMSD and pLDDT using ESMfold of raw sequences rapidly deteriorates with size for both monomers and oligomers (Fig. 4C,D; red). Consistent with prior work (Goverde et al, 2022; Verkuil et al, 2022), this suggests that optimising sequences for high AlphaFold confidence may result in adversarial examples for AlphaFold. In contrast, redesigned sequences are predicted to fold confidently using both AlphaFold (Fig. 4C,D; dark blue) and ESMfold (Fig. 4C,D; light blue), with median scRMSD < 2 Å and pLDDT > 70.

Furthermore, ADM redesign reduces the relative frequency of large hydrophobic amino acids (PHE, TRP, TYR) associated with poor protein solubility (Fig. 4E,F). Conversely, redesign results in a higher fraction of polar and charged amino acids in the protein surface (Fig. 4E) which is predicted to improve designed protein solubility. This suggests that ADM sequence redesign is indeed vital to ensure successful design of unbiased sequences as measured by scRMSD and predicted solubility.

## AlphaDesign ADM is comparable to ProteinMPNN with complementary structure preference

Next, we compared the performance of our ADM on de novo designed proteins to ProteinMPNN (Dauparas et al, 2022) a state-of-the-art model for protein sequence design. We generated 32 sequences for each protein using either ProteinMPNN or our ADM and evaluated the best sequences in terms of AlphaFold scRMSD and pLDDT for both models.

We found that on monomers, ProteinMPNN and our ADM showed very similar performance in terms of scRMSD and pLDDT of their respective generated sequences, outperforming each other on roughly 50% of de novo designed monomers (Fig. 5A). In aggregate, the percentage of successful designs was almost exactly the same for both (Fig. 5B). For oligomers, ProteinMPNN outperformed our model in terms of scRMSD in 70% of monomers, while showing comparable performance in terms of pLDDT (Fig. 5C), as well as a similar aggregate performance in terms of design success rate (Fig. 5D). This led us to believe that both models had learned complementary ways of generating protein sequences given a protein backbone.

We thus combined 16 sequences per backbone from ADM and ProteinMPNN to see if we could improve success rates by leveraging the complementarity of both methods. We found that using sequences from both ProteinMPNN and ADM improved success rates in all cases for both monomers and oligomers (Fig. 5B,D; red). This indicates that indeed our ADM can be used as an independent tool for protein sequence design to complement existing approaches.

## Generation of small protein binder designs with high AF2 success rate

Next, we applied AlphaDesign to design protein binders for a wide variety of protein targets (immune-related proteins PD-1 & PD-L1; SARS-CoV2 spike receptor binding domain (RBD); RNase A; *S. enterica*-derived retron-encoded toxin RcaT-Sen2 (Bobonis et al, 2022)). We specified an epitope (blue) on each target protein except SARS-CoV2 spike RBD, in order to demonstrate a binder design process without site-specific constraints (Fig. 6A). After benchmarking designed binders with AF, we identified computationally successful binders for each target protein. According to standard computational measures for design success (complex scRMSD < 2 Å, binder pLDDT > 70), 39.1% of designed 50 AA binders (respectively, 30.6% for 100 AA) were found to be successful (Fig. 6D,E). Comparing per-target success rates to RFdiffusion, we found increased computational success-rates for AlphaDesign binders as compared to RFdiffusion binders for all targets other than RcaT-Sen2 (Fig. 6D). Even according to more stringent criteria for computational design success proposed in (Bennett et al, 2022) with complex scRMSD < 1 Å and ipAE < 10 Å, we still recover 6% of 50 AA designs (2.7% of 100 AA designs). Evaluating only ipAE, these values increase to 21.8% for 50 AA designs (10.4% for 100 AA designs) (Fig. EV3A,B).

Additionally, to assess the plausibility of generated sequences and structures, we subjected the binder designs to all-atom

**Table 1.** Median time per design for AlphaDesign compared to other approaches.

| Method | 50 AA | 100 AA | 200 AA | 300 AA |
|---|---|---|---|---|
| AlphaDesign | 329 s | 1001.5 s | 7678 s | 35167 s |
| RFdiffusion | 157.8 s | 167.4 s | 262.2 s | 312.3 s |
| Chroma | 50.6 s | 51.1 s | 56.2 s | 66.5 s |

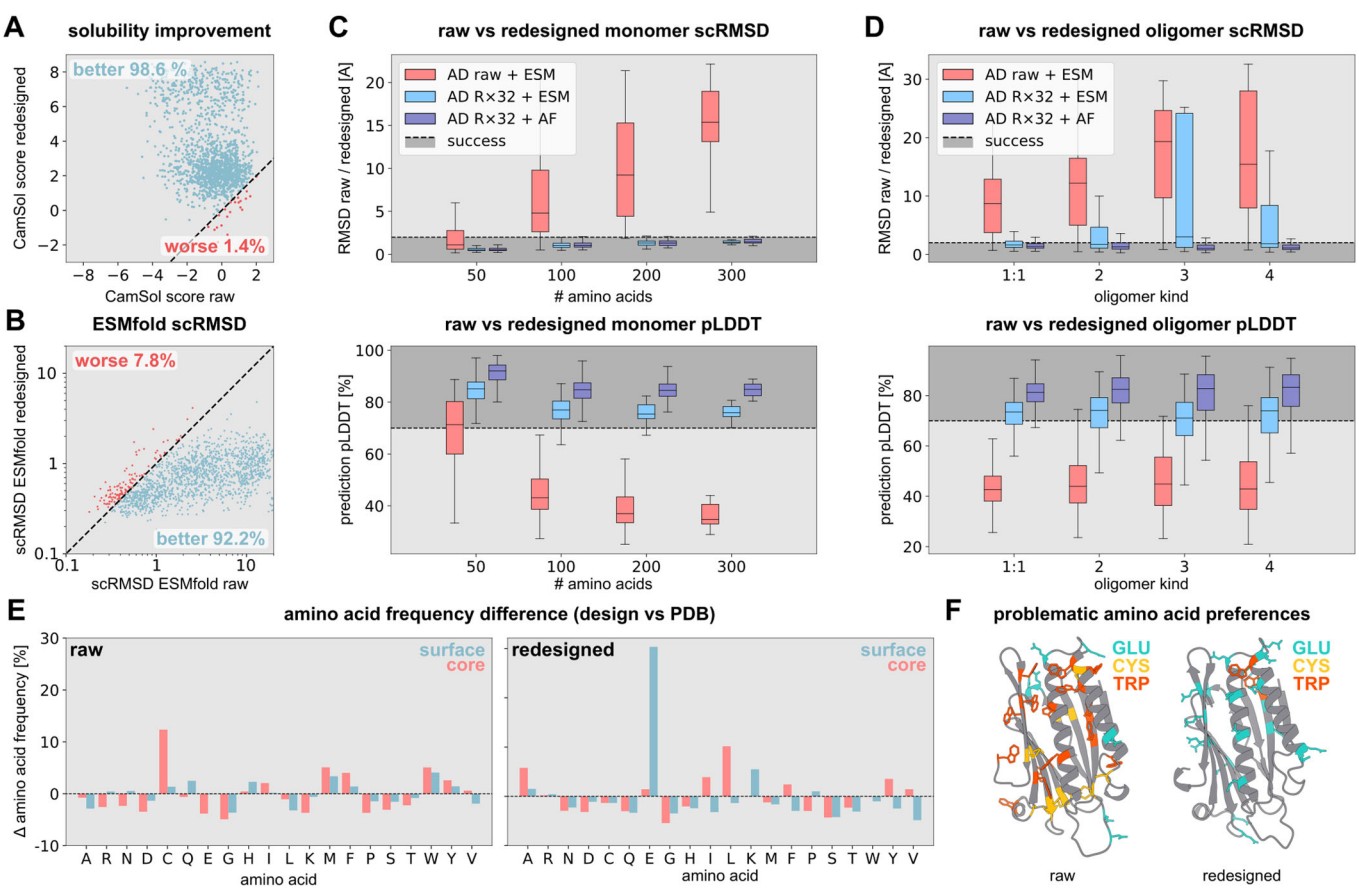

**Figure 4. Impact of sequence redesign on de novo designed proteins.**

(A) Scatter plot of CamSol (Sormanni et al, 2015) solubility scores (higher is better) for AlphaDesign candidate sequences (x-axis, N = 1681) and redesigned sequences (y-axis). Designs with increased (decreased) solubility are shown in blue (red). (B) Scatter plot of ESMfold scRMSD (lower is better) for AlphaDesign candidate sequences (x-axis) and redesigned sequences (y-axis). Designs with decreased (increased) scRMSD are shown in blue (red). (C, D) Comparison of monomer (C) and oligomer (D) design self-consistent RMSD and pLDDT for raw AlphaDesign sequences using ESMfold (AD raw + ESM), best-of-32 redesigned sequences using ESMfold and AlphaFold (AD R×32 + ESM, AD R×32 + AF). The centre line of each box in the box plot corresponds to the median, box edges to the upper and lower quartiles, and whiskers to the lowest and highest data points excluding outliers (within 1.5 times the interquartile range). AlphaDesign sample sizes for monomers, 50 AA: N = 1037; 100 AA: N = 540; 200 AA: N = 75; 300 AA: N = 29; oligomers 1:1: N = 1173; 2: N = 1267; 3: N = 319; 4: N = 251. (E) Amino acid frequency difference between candidate sequences and PDB (top), as well as redesigned sequences (bottom). Difference for surface residues is shown in blue, for core residues in red. (F) (left) Example AlphaDesign candidate with a large number of tryptophan (red), cysteine (yellow) and a small number of glutamine (blue). (right) The same protein after redesign, without cysteine and a greatly reduced number of tryptophans. Source data are available online for this figure.

molecular dynamics (MD) simulations and affinity prediction using the PRODIGY predictor (Xue et al, 2016). Binders showed low root mean square fluctuation (RMSF; Fig. 6A–C), a consistent number of interface contacts (Appendix Fig. S2A), as well as predicted affinities in the µM to nM range (Fig. EV3C). While PRODIGY was developed for affinity prediction of existing structural interfaces and may not necessarily extend to de novo designed interfaces, reaching reasonable predicted $K_D$ across a statistical ensemble of structures provides an additional check of design plausibility. This indicates that AlphaDesign produces computationally plausible binder designs, both according to AF2 prediction and molecular dynamics simulations.

## Bispecific and conformation-changing binder design

Beyond simple binder design, we generated bispecific binder designs predicted to bind different homologues of the spike

receptor binding domain (SARS-CoV2 spike RBD for the delta and omicron variants, as well as MERS spike RBD) and Retron cold anaerobic Toxin (RcaT-Sen2, Rcat-Eco1) (Figs. 6B and EV4A), as well as binder designs predicted to change conformation upon target binding (Figs. 6C and EV4B; Appendix Fig. S2B). As for conformational change designs, we were able to control the degree of conformational change by setting a threshold on the TM-score between the bound and unbound state (Fig. EV4B). Furthermore, some conformation-changing binder designs exhibit higher RMSD and RMSF in MD simulations of the monomeric state compared to the complex state (Appendix Fig. S2B). This indicates a transition from a flexible monomeric structure to a more rigid complex structure. While these design tasks yielded a much lower computational success rate, this is to be expected as the design problem is more constrained. This demonstrates the flexibility of our method, which allows for machine learning-based multi-state protein design.

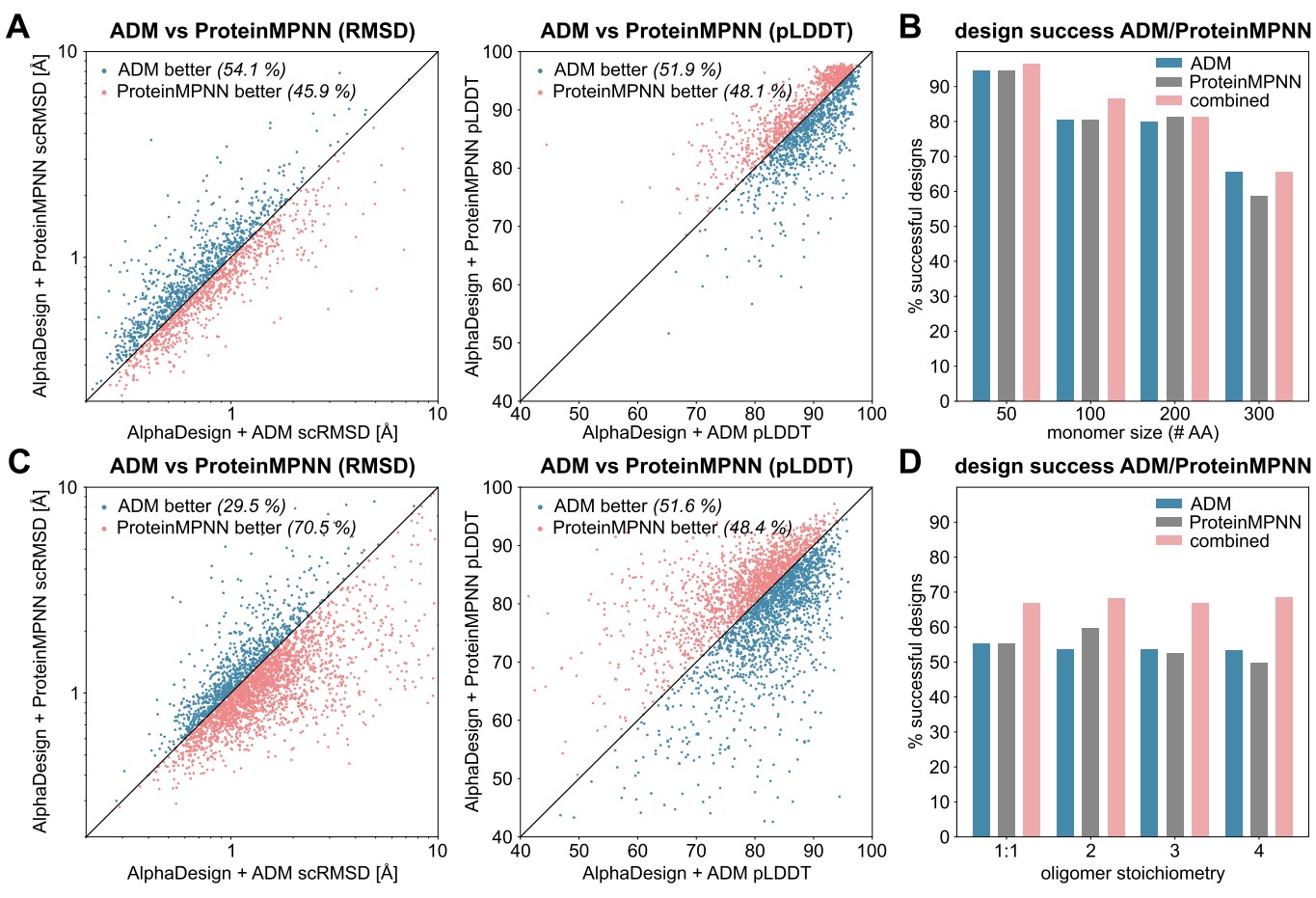

**Figure 5. Comparison of ADM and ProteinMPNN.**

(A, C) Scatter plots of scRMSD and pLDDT for AlphaDesign + ADM (x-axis) and AlphaDesign+ProteinMPNN on de novo designed monomers (A, N = 1681) and oligomers (C, N = 3010). Designs where ADM is better (worse) are marked in red (blue). (B, D) Design success rates for AlphaDesign + ADM (blue), AlphaDesign+ProteinMPNN (grey) and a combination of both (red) for monomers (B) and oligomers (D). Sample size as in Fig. 3. Source data are available online for this figure.

## De novo designed small protein binders against phage defense systems

To experimentally demonstrate our ability to design in vivo functional proteins, we generated inhibitors against RcaT (Retron cold anaerobic Toxin), a bacterial retron-encoded toxin, which is used to defend against viral infection (Bobonis et al, 2022). Upon bacteriophage infection, RcaT toxins are activated and inhibit bacterial growth to stop viral proliferation in infected cells, and ultimately spread to the population (Bobonis et al, 2022).

The RcaT family of toxins provides an essentially ideal test-bed for inhibitor design, as it comes with a variety of natural inhibitors and a readily interpretable growth-based assay for inhibitor screening (Bobonis et al, 2022). We designed binders to the active site (blue) of *Salmonella enterica*-derived RcaT-Sen2 (72 designs, Appendix Table S1) and bispecific binders for RcaT-Sen2 and its homolog RcaT-Eco1 in *E. coli* (16 designs, Appendix Table S1).

Currently, only the structure of RcaT-Eco1 has been experimentally determined (Wang et al, 2022; Carabias et al, 2024). In contrast, no experimental structures exist for RcaT-Sen2 and other RcaT homologues. Therefore, we used an AF-predicted structure as the basis for our designs. We assessed the level of RcaT inhibition

of our designs by measuring *E. coli* colony growth in the presence of both RcaT and our designed inhibitors (Fig. 7A; see Methods). Experimental measurements of RcaT inhibitor activity revealed that 19.3% of designs (17/88 total, 16/72 or 22.2% of RcaT-Sen2 designs, 13/61 or 21% at 50 AA, 5/11 or 45% at 100 AA; 1/16 or 6.2% of RcaT-Sen2/RcaT-Eco1 bispecific designs) significantly inhibit RcaT-Sen2 relative to a non-inhibiting control in vivo (Fig. 7B,C; Appendix Table S1, Appendix Fig. S3). Some designed binders (5.7%, 5/88, Appendix Table S1) inhibit RcaT-Sen2 even at leaky levels, without induction, suggesting that those are strong binders. Furthermore, a subset of designed binders also inhibit the activity of RcaT-Eco9—a homolog of RcaT-Sen2—in vivo (Fig. 7C; Appendix Table S1), showing that our method can yield robust, broad-spectrum inhibitors for closely related proteins.

While our experimental approach allows us to measure inhibitor activity, we note it does not take into account stability and expression levels of our designs. Furthermore, a subset of designs (29/88, 32%) exhibit signs of toxicity, which results in lower measured inhibitor activity.

To assess how our computational measures of design success relate to experimental success, we compared experimental blocker activity for 50 amino acid designs with scRMSD < 1.0 Å and those

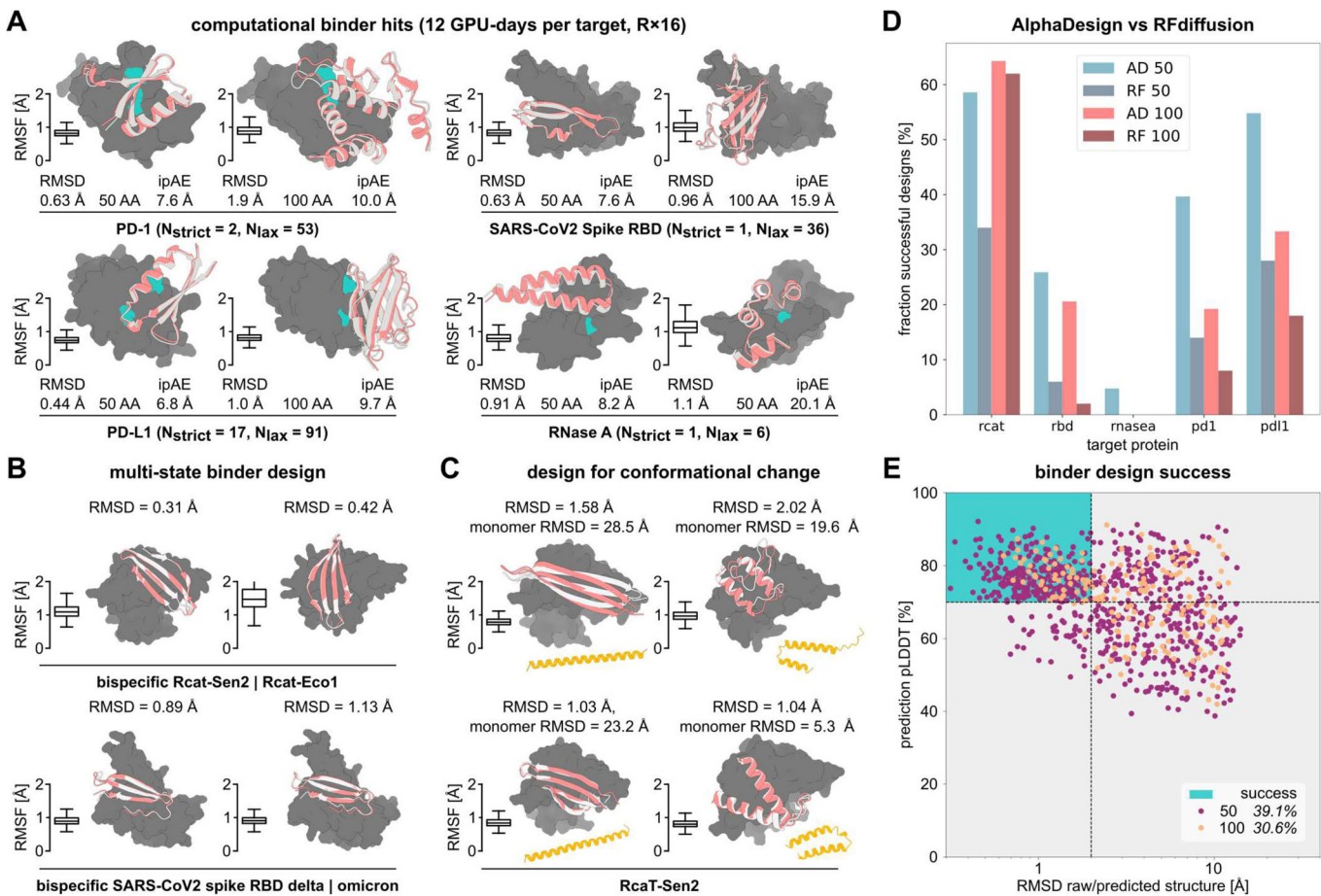

**Figure 6. Protein binder design.**

(**A**) Example binder designs for 4 target proteins (PD-1, PD-L1, SARS-CoV2 Spike RBD, RNase A). The number of successful binders according to strict (scRMSD < 1.0 Å, pLDDT > 80, ipAE < 10 Å) and lax criteria (scRMSD < 2.0, pLDDT > 70) are reported for each target protein. Target proteins are shown in dark grey, with specified binding residues for design shown in blue. Binder designs are shown in light grey, overlaid with the AlphaFold predicted binder structure in red. scRMSD and mean interface pAE are reported for each binder. The inset box plots in (**A–C**) show the complex root mean square fluctuation (RMSF) for $N = 20{,}000$ snapshots from ensemble molecular dynamics. The centre line of each box corresponds to the median, box edges to the upper and lower quartiles, and whiskers to the lowest and highest data points excluding outliers (within 1.5 times the interquartile range). (**B**) Example bi-specific binder designs for RcaT-homologues (RcaT-Sen2, RcaT-Eco1) and SARS-CoV2 Spike RBD variants (delta, omicron). Binder designs are shown in light grey, overlaid with the AlphaFold predicted binder structure in red. scRMSD is reported for each binder. Inset box plots show complex RMSF as in (**A**). (**C**) Example RcaT-Sen2 binders designed to change conformation upon toxin binding. The bound state is shown as in the preceding panels. The monomeric state is shown in orange. The RMSD between the predicted monomeric and complex states, as well as the scRMSD for the complex state are reported for each design. Inset box plots show complex RMSF as in (**A**). (**D**) Percentage of successful designs out of $N$ designs generated per target and number of amino acids for AlphaDesign (light blue: 50 AA, PD1: $N = 121$, PD-L1: $N = 135$, RcaT-Sen2: $N = 186$, SARS-CoV2 Spike RBD: $N = 112$, RNaseA: $N = 126$; light red: 100 AA, PD1: $N = 26$, PD-L1: $N = 51$, RcaT-Sen2: $N = 42$, SARS-CoV2 Spike RBD: $N = 34$, RNaseA: $N = 30$) and RFdiffusion (dark blue: 50 AA, dark red: 100 AA, $N = 50$ for all targets). (**E**) Scatter plot of redesigned sequence scRMSD, pLDDT and interface pAE for binder designs to all target proteins considered in this work. Designs are coloured by number of amino acids and percentages of successful designs are reported. The region of successful designs with scRMSD < 2.0 A and pLDDT > 70.0 is marked in blue. Source data are available online for this figure.

with scRMSD ≥ 1.0 Å. This comparison showed a significantly larger mean inhibitor activity for low scRMSD designs (t-test, $p = 0.0048$, Fig. 7D), indicating that AF scRMSD is a meaningful measure for selecting successful designs. In addition, experimentally successful binders displayed low scRMSD and ipAE, consistent with predicted binding (Fig. 7E), as well as low RMSF in molecular dynamics simulations (Fig. 7E, inset). Overall, these systematic validation experiments verify, at the phenotypic level, that our pipeline is capable of designing and selecting functional protein binders.

## Biophysical characterization of RcaT-Sen2 inhibitors

To check if our active inhibitor designs can be expressed in a soluble way and adopt the expected tertiary structure, we produced the 11 designs with the highest normalized inhibitor activity in *E. coli*. We were able to overexpress and purify 7/11 designs (Appendix Fig. S4A). Of these 7 designs, initial circular dichroism (CD) experiments indicated that one was likely unfolded in the absence of RcaT-Sen2. To validate the secondary structure content of the remaining 6 designs, we ran CD spectroscopy experiments and compared the predicted

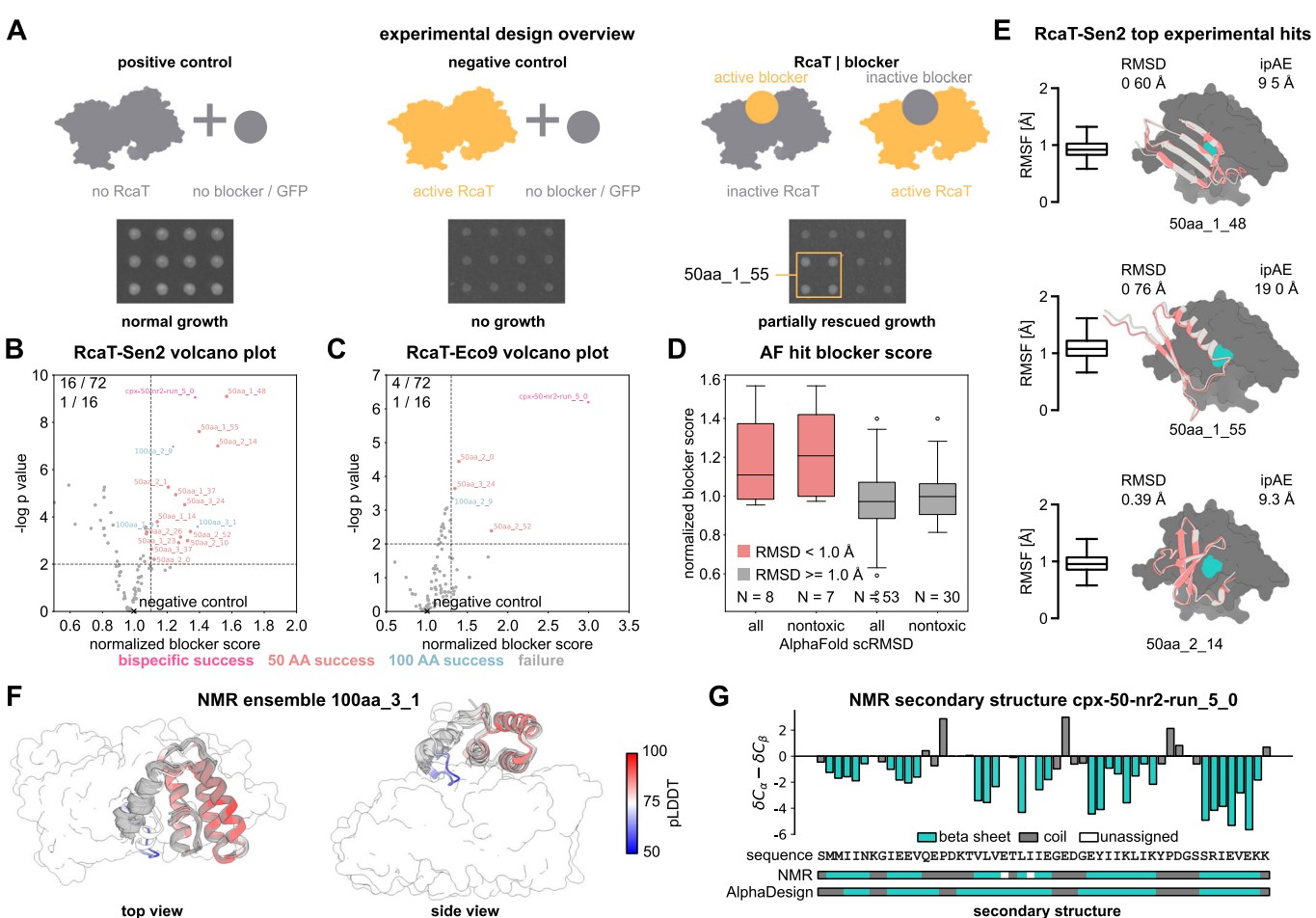

**Figure 7. RcaT experimental validation.**

(A) Schematic of the experimental setup. In the absence of active RcaT-Sen2, normal growth of E. coli colonies is expected (left). In the presence of active RcaT-Sen2 and absence of an active blocker (or presence of GFP negative control), lowered or ablated growth of *E. coli* colonies is expected (centre). In the presence of a designed blocker, RcaT activity should be partially suppressed, restoring growth (right). The lower row shows patches of plates from each condition in the TIC assay. (B, C) Volcano plot of RcaT-Sen2 (B) and RcaT-Eco9 (C) toxin inhibition activity relative to a GFP negative control, showing successful 50 AA and 100 AA designs (in red and blue, respectively), as well as successful bispecific designs (in purple). Data shown are derived from 3 biological × 2 technical replicates per condition. *p*-values were calculated using a two-sided Welch's t-test comparing blocker activity (*n* = 6 per blocker) to the activity of a GFP control (*n* = 16) (see Methods). The hit significance cutoff (*p*-value < 0.01) is shown as a dotted line. An additional effect-size cutoff is shown as a vertical dotted line (1.1 for RcaT-Sen2, 1.3 for RcaT-Eco9). 19% (17/88) of designed RcaT binders exhibit significant inhibition. (D) Box plot of RcaT-Sen2 normalised blocker scores for 50 amino acid designs with AlphaFold scRMSD < 1.0 Å (red, *N* = 8 total, *N* = 7 nontoxic) and scRMSD ≥ 1.0 Å (grey, *N* = 53, *N* = 30 nontoxic), exhibiting a significant difference in the mean with *p* = 0.0047 for all designs and 0.0036 for nontoxic designs only. Significance was tested for using a two-sided Student's t-test. The centre line of each box corresponds to the median, box edges to the upper and lower quartiles, and whiskers to the lowest and highest data points excluding outliers (within 1.5 times the interquartile range). (E) Designed and predicted structures for example inhibitors (red) of RcaT-Sen2 (dark grey) as determined using a toxin inhibition assay. RcaT-Sen2 is shown in dark grey, with the active-site-adjacent valine (the closest surface-exposed residue to the active site) used to guide binder design marked in blue. Binder designs are shown in light grey, overlaid with the AlphaFold predicted binder structure in red. scRMSD and mean interface pAE are reported for each binder. The inset box plots show the complex root mean square fluctuation (RMSF) for *N* = 20,000 snapshots from ensemble molecular dynamics. The centre line of each box corresponds to the median, box edges to the upper and lower quartiles, and whiskers to the lowest and highest data points excluding outliers (within 1.5 times the interquartile range). (F) NMR ensemble of 100 amino acid RcaT-Sen2 inhibitor 100aa_3_1 (grey; PDB 9HSA, BMRB 34973) overlaid with the predicted structure of the inhibitor (coloured) in complex with RcaT-Sen2 (white surface). The predicted structure is coloured by AF pLDDT. The first 18 amino acids of the ensemble are not shown as they were disordered. Low-pLDDT regions in the predicted structure correspond to flexible regions in the NMR ensemble. (G) Barplot of secondary chemical shifts ($\delta C_\alpha - \delta C_\beta$) for bispecific inhibitor design cpx-50-nr2-run_5_0 (BMRB 52768), together with its sequence, design secondary structure and secondary structure assigned by NMR. Bars are coloured by secondary structure (beta strand: blue, coil: grey). Source data are available online for this figure.

secondary structure content of the designs to the secondary structure content calculated from the measured CD spectra (Appendix Fig. S4B,C). We found that the secondary structure compositions of all 6 designs were similar to the predicted ones. In addition, we performed size exclusion chromatography coupled to multi-angle light scattering (SEC-MALS) measurements, to assess the oligomeric state of designs in the absence of a target, and check for possible aggregation (Appendix Fig. S5A,B). We found that none of the tested designs showed aggregation, and all 50 amino acid designs showed molecular weights consistent with the presence of dimers, while the 100-amino-acid design tested (100aa_3_1) was consistent with a monomer (Appendix Fig. S5B).

To further validate the structure of a subset of designs, we acquired NMR spectra of labelled versions of the designs 100aa_3_1 and cpx-50-nr2-run_5_0. NMR structure determination of 100aa_3_1 resulted in an ensemble closely matching the designed structure for residues 25 to 100 (Fig. 7F). Residues 1 to 24 corresponding to the N-terminus of 100aa_3_1 showed comparatively low pLDDT in the range between 50 and 70 in the designed structure and were determined to be highly flexible in the NMR ensemble (Fig. 7F). However, the C-terminal part of 100aa_3_1, which was designed to be in contact with the RcaT-Sen2 active site, was reproduced by the ensemble with RMSDs in the range of 1.45 to 2.01 Å. The first 15 residues of cpx-50-nr2-run_5_0 exhibited conformational exchange indicative of flexibility in the millisecond timescale, which precludes determination of a structural ensemble (Kleckner and Foster, 2011). However, the secondary structure obtained from NMR chemical shifts largely matches the prediction (Fig. 7G). In addition, we could verify the relative placement of $\beta$-strands in the designed structure (Appendix Fig. S6A), as well as inter-monomer contacts in the predicted dimer using long-range NOEs (Appendix Fig. S6B). This indicates that the structure of these designs at ambient temperature is indeed close to the structure generated by AD.

## Discussion

To summarise, we have developed a hybrid hallucination-diffusion approach for protein design with the versatility to enable complex, non-standard design tasks without the need for model re-training or fine tuning. Our approach compares well with diffusion models on standard tasks such as de novo monomer and oligomer design as well as the design of potent site-specific protein binders in vivo. In addition to this, it also enables multi-state protein design including generating proteins predicted to bind two related targets as well as design of proteins that are predicted to change conformation upon binding, which has not previously been realised by machine learning-based protein design methods without the use of manually designed blueprints (Praetorius et al, 2023).

We showed competitive performance of our method on computational protein design benchmarks and demonstrated its real-world applicability by designing in vivo active inhibitors to a family of bacterial phage defense proteins with previously-unknown structure. For the top inhibitor designs, we additionally verified folding and structure using CD as well as NMR structure determination. Our work is the first to demonstrate de novo design of phage defense inhibitors at this scale. Future work on de novo design of anti-phage defense proteins could help engineer phages that are resistant to multiple defense systems, facilitating future phage therapy applications.

While solely diffusion-based methods for protein design are available (Watson et al, 2023; Frank et al, 2024), our method is sufficiently flexible to design proteins according to any fitness function. Thus, we show that hallucination-based protein design methods can compete with, and provide an advantage over generative models for protein design. Future improvements in single-sequence prediction accuracy and speed of protein structure predictors are expected to have a direct impact on hallucination-based protein design. As such, our method will likely benefit from these future developments.

Despite the versatility of our method it also inherits the primary challenge of hallucination-based approaches—namely scaling poorly to larger proteins. This is in part due to the $O(N^3)$ complexity of AlphaFold and, further, due to the number of iterations required to find high-fitness sequences for larger proteins. Larger proteins require both a higher number of iterations to design and take more time per iteration, leading to even less optimal scaling behaviour. Therefore, future work could include accelerating protein structure generation by using lighter-weight structure predictors, more powerful optimisation algorithms or developing fitness-guided generative models to improve structure generation while maintaining flexibility. However, even with these computational limitations, our method is capable of designing small proteins with reasonable computational cost. Similar to other computational approaches to binder design (Watson et al, 2023; Pacesa et al, 2024), our method relies on manual target site selection and target cropping to reduce computational cost by reducing the total number of predicted amino acids. While such approaches have been successful thus far, manual target site selection and cropping has the potential to introduce design errors. Therefore, future work developing automated target site selection and cropping procedures could improve computational binder design methods. In addition, while we were able to demonstrate the design of in vivo active binders for two members of the RcaT family of bacterial toxins, it remains to be seen how well our method transfers to other target proteins. Similarly, this study is limited by the lack of high-throughput structural validation of generated structures, which we plan to address in future work.

Taken together, AlphaDesign enables rapid de novo design of small ordered protein binders while retaining the flexibility of optimizing arbitrary fitness functions. This way, AlphaDesign can be adapted to non-standard design tasks including multi-target and multi-state binder design.

## Methods

**Reagents and tools table**

| Reagent/Resource | Reference or Source | Identifier or Catalog Number |
|---|---|---|
| **Experimental models** | | |
| *E. coli* DATC | DSMZ | DSM 116187 |
| *E. coli* BW25113 | DSMZ | DSM 27469 |
| *E. coli* BW25113 *pir116* | Typas Group EMBL | |
| *E. coli* BL21(DE3) Codon Plus RIL | Agilent / Stratagene | Cat #230245 |
| **Recombinant DNA** | | |
| pJB158 | Addgene | Cat #202601 |
| pJB37 | Addgene | Cat #202602 |
| pJB117 | Typas Group EMBL | |
| pETM11-SUMO3eGFP | EMBL PEPCF | |
| **Antibodies** | | |
| **Oligonucleotides and other sequence-based reagents** | | |
| Primer JB563 (5′-aatgagggcatcgttcccac-3′) | Typas Group EMBL | |

| Reagent/Resource | Reference or Source | Identifier or Catalog Number |
|---|---|---|
| Primer JB564 (5'-cgcatacgctacttgcattacag-3') | Typas Group EMBL | |
| Primer JB565 (5'-cagtgataacggaccgcg-3') | Typas Group EMBL | |
| Primer JB566 (5'-ccgagcgttctgaacaaatc-3') | Typas Group EMBL | |
| **Chemicals, Enzymes and other reagents** | | |
| Tryptone | Sigma-Aldrich | Cat #T7293-1KG |
| Bacto yeast extract | Gibco | Cat #212750 |
| Sodium chloride NaCl | Merck | Cat #1.06404.1000 |
| Glucose | Merck | Cat #1.04074.1000 |
| Lactose | Merck | Cat #1.07657.1000 |
| Magnesium sulfate heptahydrate $MgSO_4.7H_2O$ | Merck | Cat #1.05886.1000 |
| DAP | Sigma-Aldrich | Cat #D1377 |
| Spectinomycin | Sigma-Aldrich | Cat #S9007 |
| Tetracycline hydrochloride | Sigma-Aldrich | Cat #C4881 |
| Kanamycin | Roth | Cat #T832.2 |
| Chloramphenicol | BioChemica | Cat #A1806,0025 |
| IPTG | Carl Roth | Cat #2316.3 |
| l-Arabinose | Sigma-Aldrich | Cat #A3256 |
| Gentamicin | Sigma-Aldrich | Cat #G1264 |
| BsaI-HF-V2 | NEB | Cat #R3733L |
| GoTaq Green 2x master mix | Promega | Cat #M7122 |
| Glycerol | VWR | Cat #1.04091.1000 |
| Lysozyme | Sigma-Aldrich | Cat #62971 |
| SmNuclease | EMBL PEPCF | |
| cOmplete™, EDTA-free Protease Inhibitor Cocktail | Sigma-Aldrich | Cat #COEDTAF-RO |
| Tris (Trizma base) | Sigma | Cat #T1503 |
| Imidazole | Merck | Cat #1.04716.1000 |
| $D_2O$ | Sigma-Aldrich | Cat #151882 |
| Ammonium chloride ($^{15}N$) | Merck | Cat #299251 |
| D-glucose ($^{13}C$) | Merck | Cat #389374 |
| Ni-NTA beads | Qiagen | Cat #30210 |
| Poly-Prep Chromatography Columns | Bio-Rad | Cat #7311550 |
| $His_6$-tagged SenP2 protease | EMBL PEPCF | |
| 1 mL Protino Ni-NTA column | Macherey-Nagel | Cat #745410.5 |
| Spectra/Por 3 Dialysis Tubing, 3.5 kD MWCO | Repligen | Cat #132720 |
| Amicon Ultra-4 3 kDa MWCO centrifugal filter | Merck-Millipore | Cat #UFC800308 |
| Superdex™ 75 10/300 GL column | Cytiva | Cat #17517401 |
| Superdex™ 200 Increase 5/150 GL gel-filtration column | Cytiva | Cat #28-9909-45 |
| NMR tubes | Norell | Cat #ST500-7 |

| Reagent/Resource | Reference or Source | Identifier or Catalog Number |
|---|---|---|
| **Software** | | |
| Adobe Photoshop 2023 (v.24.7.0) | Adobe | |
| R v.4.3.1 | | |
| Astra 8.2.0 | Wyatt Technology | |
| NMPipe | Delaglio et al, 1995 | |
| Cara | http://cara.nmr.ch | |
| NMRViewJ | Keller, 2005; Johnson and Blevins, 1994 | |
| CYANA 3.98.15 | Güntert, 2009 | |
| TALOS+ | Shen et al, 2009 | |
| AMBER | https://ambermd.org/ | |
| RStudio v. 2023.06.1 + 524 | Posit | |
| **Other** | | |
| Singer pinning robot Rotor HDA | Singer Instruments | Cat #ROT-001 |
| Humid incubator MCO-80IC | Sanyo | MCO-80IC |
| PlusPlates | Singer Instruments | Cat #PLU-003 |
| Canon EOS Rebel T3i camera | Canon | |
| Microfluidizer M-110L FluidProcessor | Microfluidics | |
| Optima L-100 XP Ultracentrifuge | Beckman Coulter | |
| ÄKTA Pure chromatography system | Cytiva | |
| Jasco J-815 CD spectrophotometer | Jasco | |
| Agilent 1260 Infinity II HPLC system | Agilent | |
| MiniDAWN and Optilab | Wyatt Technology | |
| Bruker Avance III HD 700 MHz | Bruker | |
| Bruker Avance III HD 900 MHz | Bruker | |
| Bruker Avance III 600 MHz | Bruker | |

## Methods and protocols

We frame protein design as a search problem to find the set of protein sequences for which a certain fitness function exceeds a fixed threshold. To take into account both sequence properties and all-atom protein structure, we integrate AF (Jumper et al, 2021) into our fitness functions to provide high-quality structure prediction and measures of prediction confidence. We combine this with state-of-the-art validation using Rosetta ab initio structure prediction (Leaver-Fay et al, 2011) and all-atom molecular dynamics simulations.

## AlphaDesign pipeline

The AlphaDesign pipeline consists of three separate steps: Candidate structure generation, sequence redesign and computational validation. In the first step, we use an evolutionary algorithm (Sinai et al, 2020) to identify sequences with high-confidence AF predictions conforming to a design goal (Methods, "Candidate structure generation with AlphaFold"; Fig. 1, red). In the second step, we use an autoregressive diffusion model (ADM) (Hoogeboom et al, 2022) to generate optimised sequences for each candidate structure (Sec. 4.2.5; Fig. 1, blue) and discard candidates for which no optimised sequence can be found (Methods, "Sequence redesign with autoregressive diffusion models"). Finally, we predict the structures of these optimised sequences using AF and ESMfold, discarding designs with high RMSD to the original design, or low pLDDT (Methods, "Computational benchmarks"). The remaining candidates are codon-optimised and can be ordered as gene fragments for experimental validation (Methods, "Codon optimisation").

## Fitness functions for protein design tasks

In order to frame protein sequences and structure design as a search or optimisation problem, we need to specify a fitness function which encapsulates the end goal of a given design task. AlphaFold pLDDT and pAE for a prediction correlate with protein structure quality (Mirdita et al, 2022). Therefore, optimising for high pLDDT and low pAE should provide a good prior for generating native-like protein structures. Interface pAE (ipAE) correlates with the probability of protein–protein interaction (Evans et al, 2021). Optimising for low ipAE between two protein monomers allows us to design protein–protein interactions. In combination with additional confidence and geometric constraints (Table 2), this allows us to specify fitness functions for de novo monomer, oligomer ($L_{denovo}$) as well as targeted protein binder design ($L_{binder}$). We can also search for sequences for which AF predicts different monomeric and oligomeric states by designing a corresponding fitness function ($L_{change}$).

## Candidate structure generation with AlphaFold

We repurpose AF (Jumper et al, 2021) as a generative model of protein backbones. By searching for amino acid sequences with high-confidence predictions we can extract native-like protein structures from AlphaFold (Fig. 1A). To automate this search we use an evolutionary algorithm to continually update a pool of amino acid sequences. The sequence pool is initialised uniformly at random from a set of allowed amino acids. In this study, we exclude cysteine as it could interfere with protein expression (Wicky et al, 2022; Frank et al, 2024; Kong and Guo, 2014). AF predicts structures and confidence scores (predicted local distance difference test, pLDDT; predicted aligned error, pAE) for these sequences. We then combine these confidence scores into a fitness function with additional constraints on protein sequence, structure and function (Fig. 1B). The evolutionary algorithm then updates the sequence pool. In this work, we use an evolutionary algorithm following (Sinai et al, 2020) to optimise the fitness function. However, other gradient-free or gradient-based optimisers can be substituted as desired. Throughout optimisation a protein changes

its structure and both local and global confidence measures increase. This process is repeated until a sequence in the pool reaches a user-defined fitness threshold (see Supporting Methods, "AlphaDesign design loop pseudocode"). The specified fitness function (Fig. 1B) dictates the structure and function of protein structures generated this way.

## Fixing protein structures using templates

To enable binder design without requiring MSA inputs for target proteins, we provide AF with a template structure for proteins whose structure should remain fixed throughout the design process. To reduce computation as much as possible, we may crop the structures of target proteins around the desired binding site. We then feed this cropped sequence and structure as a template into AF. This keeps computational cost fixed for binder design to arbitrary size proteins.

## Autoregressive diffusion model training

To generate improved sequences for structure candidates recovered from AF, we trained a conditional autoregressive diffusion model (ADM) (Hoogeboom et al, 2022) on protein sequences and structures in the PDB. Conditional ADM learn to recover the amino acid sequence of a protein given its structure by optimising the following loss (Hoogeboom et al, 2022):

$$L_{ADM}(\theta) := E_{s,x \sim p_{data}, s' \sim p_{mask}(s'|s)} [\log p_\theta(s|s', x)]$$

where $p_\theta$ denotes the output distribution of protein sequences from the model, $p_D$ denotes the data distribution, $s$ denotes a protein sequence, $x$ a protein structure and $p_{mask}(s'|s)$ denotes a distribution of masked amino acid sequences $s'$ conditioned on a ground-truth sequence $s$. We chose $p_{mask}$ to be the uniform distribution over masked sequences. That is, the number of masked positions was sampled uniformly at random as were the masked positions. We parameterised $p_\theta(s|s', x)$ as a product $\prod_i p_\theta(s_i|s', x)$ of independent categorical distributions for each amino acid as customary for masked sequence modelling (Hoogeboom et al, 2022).

We based the architecture of our ADM on the message-passing blocks of ProteinMPNN (Dauparas et al, 2022). Our model consists of 6 MPNN blocks with 30 nearest neighbours, 128 node features and 128 edge features. Before passing the protein backbone to the MPNN, we add Gaussian noise with mean $\mu = 0.0$ and standard deviation $\sigma = 0.3\,\text{Å}$ to all atom positions. In contrast to ProteinMPNN, node features were initialised with the embedding of the masked amino acid sequence and the model was trained to directly predict independent probability distributions for each masked amino acid. This made it easier to implement complex dependencies between amino acids in homooligomers or repeat proteins during sampling.

The training set consisted of all protein structures from the PDB (Berman et al, 2000) with a resolution better than 3.5 Å deposited before January 2021. Validation and test sets were drawn from structures in the PDB deposited before January 2022 with less than 10% sequence identity to the training set. The training set was clustered at 40% sequence identity. Structures were sampled from the training set with a probability inversely proportional to cluster

size without replacement. Structures were accumulated into batches until adding an additional chain would increase the batch size beyond 10k amino acids. Batches were zero-padded to 10k amino acids and padded amino acids were masked out during training. For simplicity of implementation the loss was averaged over all masked amino acids in a batch, as opposed to averaging per sequence.

The model was trained for 250k steps using Adam (Kingma and Ba, 2014) with hyperparameters $\beta_1 = 0.9$ and $\beta_2 = 0.98$ and linear warm-up to a final learning rate of $1e-3$ over the first 10k steps. These hyperparameters were chosen following ProteinMPNN. During training, overfitting was assessed by monitoring perplexity of the model on the validation set. As we did not observe an increase in the validation loss across the entire 250k steps of mode training, we selected the final checkpoint for further evaluation.

## Sequence redesign with autoregressive diffusion models

We redesign sequences of candidate structures using an ADM starting from a fully masked sequence. For binder design, only the sequence of the binder was masked, while the target sequence was kept intact. At each step of redesign, we sample a random masked position, compute the probability distribution at that position, set the probability of forbidden amino acids to zero, re-scale it by an inverse temperature of 10 and sample an amino acid. If the sampled position has any sequence constraints associated with it, we also set the sampled amino acid at all tied positions. We repeat this process until no more masked amino acids remain. This way, we sampled 100 sequences per candidate structure and ranked them by their likelihood under the model (see Supporting Methods, "Sequence design pseudocode").

## Computational benchmarks

To computationally verify the folding of designed proteins, we predict their structures using AlphaFold and ESMfold in single-sequence mode with 4 recycling iterations. We compute the mean $pLDDT$, $pAE$, interface $pAE$ ($ipAE$) and $pTM$ for those structures. We then compute the self-consistent RMSD (scRMSD) and self-consistent TM-score (scTM) between the original designed structure (using AlphaDesign or RFDiffusion) and predicted structures of the corresponding designed sequences. For binder design, we input a template for the target protein to AlphaDesign, and initialise recycling with the designed structure of the complex following (Bennett et al, 2022). For multi-state designs (conformation-changing proteins, bispecific binders), we run these benchmarks once for each state (bound/unbound, binding target 1/2) and combine the results. In all benchmarks involving sequence redesign, we report results for the best redesigned sequence, selected by first applying all thresholds for design success and then sorting by scRMSD. For comparison with RFDiffusion (Fig. 1E,F, bottom row), we use 8 redesigned sequences (randomly sampled from 100) to be consistent with the evaluation reported in (Watson et al, 2023). For comparison between raw and redesigned sequences (Fig. 1E,F, top row) we use 32 redesigned sequences (randomly sampled from 100). For protein binder design, we use 100 random redesigned sequences, as sequence generation and benchmarking are generally cheap compared to candidate generation. Unless stated otherwise, we consider a designed protein *computationally*

*successful* if it fulfils $pLDDT>70$, $scRMSD<2$ Å. For binder design, we additionally consider a stricter threshold of $pLDDT>80$, $scRMSD<1$ Å and $ipAE<10$ Å following (Bennett et al, 2022). Benchmark code is available at github.com/mjendrusch/novobench.

## ADM benchmarks

To evaluate the suitability of our autoregressive diffusion model for de novo protein design, we generated 32 sequences for each monomer in a set of 1688 of monomers of 50 to 300 amino acids generated by optimising $L_{denovo}$ (Table 2), evaluated their scRMSD, pLDDT and success rate using AlphaFold and compared the results to 32 sequences generated using ProteinMPNN (Dauparas et al, 2022).

## RFDiffusion benchmark protein generation

To compare our method to RFDiffusion, we generated monomers of sizes 50 AA, 100 AA, 200 AA and 300 AA as well as heterodimers with 50 AA monomers, homooligomers (dimers to tetramers) with 50 AA monomers. For AlphaDesign, we generate candidate structures over 4 days on two Nvidia 3090 GPUs, resulting in 1688 total monomers and 3010 total oligomers. We then generated 100 sequences for each structure using our ADM. For comparison with RFDiffusion, we selected 8 of these redesigned sequences uniformly at random. For RFDiffusion, we generated 500 structures for each size of monomers and 500 structures for each type of oligomers. Monomers and heterodimers were generated using RFDiffusion's default settings, while homodimers were generated with cyclic symmetry enabled. Following (Watson et al, 2023), we then sampled 8 sequences for each generated structure using ProteinMPNN (Dauparas et al, 2022). For all generated structures and their corresponding sequences, we then predicted pLDDT, pAE and scRMSD as described above. Designed sequences, structures and benchmark scores were deposited on Zenodo (https://doi.org/10.5281/zenodo.15208893).

## AlphaDesign sample size

The number of backbones generated in all benchmarks for AlphaDesign depends on the difficulty of the protein design task. More complex design tasks (e.g. multi-state design) as well as design tasks result in a lower number of generated backbones. Similarly, a larger number of amino acids to design will result in a smaller number of backbones generated. As all experiments in this study operate on randomly generated and algorithmically selected protein sequences and structures, additional blinding or randomization is likely not required to mitigate bias.

## RcaT inhibitor design

We predicted the structure of RcaT-Sen2 (Bobonis et al, 2022) using ColabFold with 96 recycling steps and early stopping with a tolerance of 0.1 Å (Mirdita et al, 2022). We then extracted a 159 amino acid crop around the putative active site for RcaT-Sen2 and RcaT-Eco1 as a template for fixed-target binder design. Binder design was constrained to generate binders around V106 for RcaT-Sen2 and T105 for RcaT-Eco1. We ran AlphaDesign to generate

**Table 2. Fitness function components used in this study.**

| Fitness component | Expression | Description |
|---|---|---|
| *Confidence* | | |
| $L_{pAE}(X)$ | $1 - \frac{mean(pAE(X))}{pAE_{max}}$ | Mean normalised pairwise confidence of $X$ |
| $L_{ipAE}(X, Y)$ | $1 - \frac{mean(ipAE(X,Y))}{pAE_{max}}$ | Mean normalised interface confidence of $X$ and $Y$ |
| $L_{pLDDT}(X)$ | $mean(pLDDT(X))$ | Mean pLDDT of $X$ |
| $L_{conf}(X)$ | $\frac{L_{pAE}(X)+L_{pLDDT}(X)}{2}$ | Predicted confidence fitness function of $X$ |
| $L_{bind}(X|T;\alpha)$ | $\alpha \cdot L_{ipAE}(X, T) + (1-\alpha) \cdot L_{conf(X)}$ | Interaction confidence fitness function for a binder $X$ against a target $T$ |
| *Geometry* | | |
| $L_{TM}(X, X')$ | $\max_j mean_i(f(AE_{ij}(X,X'))); f(d) := \frac{1}{1+\frac{d}{d_0}};$ $d_0 = 1.24(\max(length(X), 19) - 15)^{\frac{1}{3}} - 1.8$ | Approximate template matching score of $X$ and $X'$ |
| $L_{site}(X|T;r,s)$ | $\max(r, \min_i(d(X_i, T_s))) - r$ | Thresholded minimum distance of amino acids in binder $X$ to the target site $T_s$ on $T$ |
| $L_R(X;R)$ | $\frac{\max(R, R_g(X)) - R}{R}$ | Thresholded constraint on the radius of gyration $R_g(X)$, |
| *Combined* | | |
| $L_{denovo}(X^1,...,X^N; \alpha = 0.7)$ | $2 \cdot (\alpha \cdot L_{conf}(X^1 : ... : X^N) + (1-\alpha) \cdot \frac{1}{N}\sum_i L_{conf}(X^i)) - L_R(X^1 : ... : X^N; R_g)$ | De novo design fitness function used in this work |
| $L_{change}(X, X : Y;\alpha)$ | $\alpha \cdot L_{conf}(X : Y) + (1-\alpha) \cdot L_{conf}(X) - ReLU(L_{TM}(X, X : Y) - TM_{max})$ | Conformational change fitness function used in this work. $X_{X:Y}$ denotes the structure of $X$ in the complex $X{:}Y$ |
| $L_{binder}(X|T;s)$ | $L_{bind}(X|T;0.7) - L_{site}(X|T;6\text{Å},s) - L_R(X)$ | Binder design fitness function used in this work |

Fitness functions are composed from multiple confidence measures (pAE, interface pAE, pLDDT) (Jumper et al, 2021) as well as geometric constraints on the predicted structure (radius of gyration $R_g$, maximum distance between pairs of residues). Confidence fitness components provide a prior that ensures generated structure candidates are native-like ($L_{conf}$) and exhibit desired protein–protein interactions ($L_{ipAE}$). Geometry-based fitness components constrain generated structure candidates to further conform to a desired topology and geometry.

candidate structures of 50 and 100 amino acids with fitness $L_{binder}(X|RcaT-Sen2; active-site)$, 2 recycling steps, suboptimality 0.1, population size 10. We generated 100 sequences per candidate structure using our autoregressive diffusion model, keeping the target sequence fixed. We kept the highest-likelihood sequence and discard candidates with $\frac{1}{N}\sum_i^N \log p_\theta(s_i|x) < -0.1$ nats. We re-predicted candidate structures with AF at 4 recycling steps, discarding candidates with $pLDDT \cdot pTM < 0.5$. We predicted sequence-based candidate solubility scores with CamSol using bulk sequence prediction on the CamSol Intrinsic webserver at pH = 7 (Sormanni et al, 2015). We then discarded candidates with score <1. We codon-optimised the remaining candidates for expression in *E. coli*. After RcaT binders were designed and benchmarked on cropped RcaT structures, we used AlphaFold to predict the structure of the complex between each designed RcaT binder and the full RcaT-Sen2 by providing a template for the cropped structure, as well as a template for the full RcaT-Sen2 structure that we previously predicted. We used these un-cropped complexes for all downstream molecular dynamics analyses.

## Codon optimisation

Designed amino sequences were reverse-translated and codon optimised using the DNA Chisel package (Zulkower et al, 2020). GC content was constrained to lie between 35% and 65%, sequences containing pentanucleotide repeats, rare codons and hairpin-forming sub-sequences were excluded. Sequences were optimised to match native *E. coli* codon usage and reduce the amount of repeat codons.

## Bacterial strains, plasmids and growth conditions

Bacteria were grown in lysogeny broth (LB; 10 g/L of tryptone, 5 g/L of yeast extract, and 5 g/L of NaCl) or on LB Lennox agar (2%) plates. *E. coli* strain DATC (DAP-Auxotroph Transformation Conjugation) (DSM 116187) was used to conjugate the designed blockers cloned in the pJB158 plasmid (Addgene ID 202601) into the BW25113 *pir*-116 strain transformed with plasmid pJB37 or pJB117 (Bobonis et al, 2022). BW25113 *pir*-116 was constructed by replacing the $\beta$-glucoside locus (*bgl*) with *pir*-116 in *E. coli* BW25113 (DSM 27469), which allows the R6K-containing pJB158 vectors to replicate in high-copy numbers (~250 copies/cell) (Metcalf et al, 1994). Antibiotics, auxotrophies, and inducers used were Spectinomycin (100 μg/mL), Tetracycline (10 μg/mL), Gentamicin (10 μg/mL), Diaminopimelic acid (0.3 mM), Isopropyl-$\beta$-D-thiogalactopyranosid (0.1 mM), and L-arabinose (0.2%).

## Construction of the genetic library of RcaT blockers

DNA sequences encoding the designed RcaT-Sen2 blockers were synthesized (IDT) and cloned through Golden Gate Assembly (Engler, 2008) using BsaI-HF-V2 (NEB) into the IPTG-inducible pJB158 vector (Addgene: #202601). Each sequence contained Golden Gate-compatible overhangs at the 5′ (5′-gcgagggtctcggcat-3′) and 3′ (5′-taagcgagacccagatc-3′) end. Golden gate reactions were performed in 96-well PCR plates in a thermocycler (30 cycles of 2 min at 37 °C and 5 min at 16 °C, followed by 10 min at 60 °C, and 20 min at 80 °C). Chemically competent *E. coli* DATC cells were transformed with 5 μL of Golden Gate reactions. Desired transformants were selected based

on lack of GFP fluorescence using blue light screening and further via colony PCR using GoTaq Green polymerase (Promega) with primers JB563 (5′-aatgagggcatcgttcccac-3′) and JB564 (5′-cgcatacgctacttgcat-tacag-3′). Transformants carrying correctly-sized inserts (as judged by colony PCR) were validated by Sanger sequencing using primers JB565 (5′-cagtgataacggaccgcg-3′) and JB566 (5′-ccgagcgttctgaacaaatc-3′), arrayed into 96-well plates, grown overnight in LB with 0.3 mM DAP and 10 µg/mL tetracycline and frozen at −80 °C after adding glycerol (12.5% final concentration). Per blocker construct, two separate transformants were selected (biological replicates).

## Toxin-inhibition conjugation (TIC) procedure

The TIC assay was done as described in Bobonis et al (2024). Briefly, TIC is a high-throughput genetics screen to identify inhibitors of TA systems. Arrayed donor libraries carrying mobilizable gene-overexpression plasmids (the designed RcaT-Sen2 blockers) were conjugated on recipient bacteria expressing a toxin plasmid (RcaT-Sen2) and colony fitness is measured on plate in presence/absence of the inducers. TA inhibitors should increase strain fitness in the presence of the toxin, by inhibiting the toxin. The TIC assay was adapted from this protocol (Bobonis et al, 2024).

By using the Singer ROTOR pinning robot, two 96-well glycerol stock plates containing the blocker library in the E. coli DATC donor strain (two biological replicates) were pinned from liquid glycerol stocks to LB DAP Tetracycline plates (96-density long-pin Singer RePads), re-arrayed into 384-colony arrays (leading to two biological with two technical replicates of the library in one 384-array), and grown overnight at 37 °C in a humid incubator. The conjugation recipient strain (E. coli BW25113 pir-116) carrying the p-RcaT-Sen2 plasmid (pJB37) or the p-RcaT-Eco9 plasmid (pJB117) was grown overnight in LB spectinomycin in three biological replicates, and 200 µL of culture (OD 595 nm = 0.5) was spread on LB DAP Singer plates using glass beads. Plates with recipient lawns were incubated in a non-humid incubator at 37 °C for 1 h to dry. Afterwards, the blocker library was pinned on top of the recipient lawns in three biological replicates, using 384 short-pin Singer RePads. Plates were incubated for 8 h in a humid incubator at 37 °C for conjugation. Next, cells were pinned from the conjugation plates onto LB plates without DAP and with Spectinomycin and Tetracycline using 384 short-pin Singer RePads, to select for transconjugants which carry both recipient (pJB37 or pJB117) and donor (blocker libraries) plasmids. Transconjugants were grown for 16 h at 37 °C before being pinned again on double selection plates for another round of selection. Plates from the second round of double selection were incubated for 8–9 h at 37 °C, and then each biological replicate was pinned using 384 short-pin Singer RePads into double-selection LB experimental plates. Experimental plates contained either no inducer, only arabinose (to induce toxin expression), only IPTG (to induce blocker expression), or both inducers.

Experimental plates were incubated for 10 h at 37 °C, and imaged using a Canon EOS Rebel T3i camera (S&P robotics).

## TIC data analysis

Bacterial colony opacity for each strain was quantified using the image analysis tool Iris (Kritikos et al, 2017), and used as a proxy for bacterial fitness. To increase background contrast for colony detection by Iris, plate images were converted to grayscale and contrast enhanced by a value of 12 on Adobe Photoshop 2023 (v.24.7.0). Colonies with opacity equal to 0 were considered pinning misses and were removed. As previously reported (Baryshnikova et al, 2010), colonies on the outermost plate edge had systematically higher opacity values, which were multiplicatively corrected to the median opacity of the rest of the plate. To determine the effect of each blocker in each condition as compared to non-blocking GFP controls and to compare between same-condition plates, the colony opacity of each blocker-carrying strain was divided per plate by the mean opacity of the GFP-carrying controls ($n = 32$), deriving normalised fitness values for each blocker ($n = 12$), which were then averaged across the two technical replicates. Each blocker's effect was then quantified as the ratio between fitness values upon toxin and blocker induction (double-induction plates with arabinose and IPTG) and upon toxin-only induction (arabinose only). Significance was assessed through a two-sided Welch's t-test comparing the effect of each blocker ($n = 6$) and GFP ($n = 16$). A cumulative effect size was derived for each blocker as the mean of its effect divided by the effect of GFP, distinguishing blockers from non-blockers, i.e. neutral (effect size = 1) or aggravating the toxin effect (effect size < 1). Leaky blockers were defined as fitness upon toxin-only expression higher than two standard deviations from the mean of GFP controls in the same condition. Toxic blockers were defined as fitness upon blocker-only expression lower than two standard deviations of GFP controls in the same condition. Hits fulfilled a significance cutoff ($p$-value < 0.01) and an effect size cutoff of 1.1 for RcaT-Sen2 (1.3 for RcaT-Eco9). Data analysis was implemented with R v.4.3.1 (R Core Team, 2020) and RStudio v. 2023.06.1 + 524 (Posit Team, 2023). Normalized opacity measurements can be found in Appendix Fig. S6.

## Small-scale expression and purification

E. coli BL21(DE3) Codon Plus RIL cells (Stratagene) were freshly transformed with expression plasmids (pETM11-SUMO3eGFP (Scholz et al, 2013)) encoding $His_6$ and Sumo3-tagged, codon-optimized, constructs for the top 11 RcaT-Sen2 inhibitor designs by normalized blocker activity. 50 mL cultures were grown in TB-FB medium supplemented with 2 mM $MgSO_4$, 0.05% glucose, 1.5% lactose, 30 µg/mL Kanamycin and 34 µg/mL Chloramphenicol. The E. coli cultures were grown at 37 °C until the OD600 0.6–0.8, after which the temperature was reduced to 18 °C. After overnight expression at 18 °C, the cultures were harvested by centrifugation (10 min, 3200 × g, 4 °C) and the pellets were flash-frozen in liquid nitrogen and stored at −80 °C until the start of the protein purification.

The cell pellets were resuspended in 4 mL lysis buffer (50 mM Tris pH 7.5; 1 M NaCl; 20 mM imidazole; 5 mM; 1 mg/mL lysozyme; 25 ng/µL SmNuclease; cOmplete™, EDTA-free Protease Inhibitor Cocktail (Sigma-Aldrich)). Cells were lysed by 4 consecutive freeze-thaw cycles, followed by centrifugation (15 min, 17,000 × g, 4 °C). After centrifugation, the supernatant was incubated with 125 µL Ni-NTA beads (Qiagen) pre-equilibrated with wash buffer (50 mM Tris pH 7.5; 300 mM NaCl; 10% glycerol; 20 mM imidazole) and the samples were rotated for 1 h at 4 °C. After this binding step, the Ni-NTA beads were transferred into an empty Poly-Prep column (Bio-Rad), washed 3 times with 4 mL wash buffer and eluted with 300 µL elution buffer

(50 mM Tris pH 7.5; 300 mM NaCl; 10% glycerol; 250 mM imidazole). The total lysate, the soluble fraction after centrifugation and the elution fraction of all samples were analysed via SDS-PAGE.

To assess solubility in the absence of the Sumo3-tag, elution fractions were subsequently incubated with 5 µg of $His_6$-tagged SenP2 protease for 2 h on ice. After the digest, samples were centrifuged (15 min, $17,000 \times g$, 4 °C) and the supernatants analysed via SDS-PAGE.

## Expression and purification for downstream analyses

Larger scale protein production experiments were performed for constructs that could be successfully expressed and purified in a soluble way in the small scale tests (Appendix Fig. S3A). The expression plasmids were again freshly transformed into *E. coli* BL21(DE3) Codon Plus RIL cells (Stratagene). Precultures were grown overnight at 37 °C in LB medium supplemented with 30 µg/mL Kanamycin and 34 µg/mL Chloramphenicol. The next day, 20 mL preculture was added to 1 L of $^{15}$N M9 minimal medium containing 0.5 g $^{15}NH_4Cl$ (water 858 mL; $Na_2HPO_4$ 6.78 g; $KH_2PO_4$ 3 g; NaCl 0.5 g; 100x trace elements 10 mL; Thiamin (50 mg/mL) 0.6 mL; glucose (20%) 20 mL; $MgSO_4$ (1 M) 2 mL; $^{15}NH_4Cl$ 0.5 g; $CaCl_2$ (1 M) 100 µL; total of 1 L) or 1 L of $^{13}$C $^{15}$N M9 minimal medium containing 2 g $^{13}$C-glucose and 0.5 g $^{15}NH_4Cl$ (water 858 mL; $Na_2HPO_4$ 6.78 g; $KH_2PO_4$ 3 g; 0.5 g; 100x trace elements 10 mL; Thiamin (50 mg/mL) 0.6 mL; $^{13}$C-glucose 2 g; $MgSO_4$ (1 M) 2 mL; $^{15}NH_4Cl$ 0.5 g; $CaCl_2$ (1 M) 100 µL; total of 1 L) in 5 L culture flasks. The 20 mL preculture was centrifuged (15 min, $1200 \times g$) and resuspended in minimal medium before addition to the culture flasks. The expression cultures were grown at 37 °C until OD600 0.6–0.8 and then induced with 0.5 mM IPTG. After induction, the temperature was reduced to 20 °C for overnight cultivation. The cells were harvested by centrifugation (20 min, $5000 \times g$), flash-frozen in liquid nitrogen and stored at −20 °C until the start of the purification.

Cell pellets from 1 L *E. coli* cultures were resuspended in 40 mL lysis buffer. The cells were lysed by 5 passages through a microfluidizer device, followed by centrifugation (30 min, 35,000 rpm, 4 °C, Ti45 rotor) in an Optima L-100 XP Ultracentrifuge (Beckman Coulter). After centrifugation, the supernatant was loaded at ca. 1 mL/min on a 1 mL Protino Ni-NTA column (Macherey Nagel) pre-equilibrated with wash buffer using a peristaltic pump at 4 °C. After this binding step, the Ni-NTA columns were washed with 15 mL wash buffer and attached to an ÄKTA system for elution. The total lysate (TL), the soluble fraction after centrifugation (supernatant SN) and the elution fraction (Elu) of all samples were analysed via SDS-PAGE. As 50aa_2_14 and 50aa_2_10 showed some precipitation in the elution fractions, the samples were centrifuged (15 min, 4500 rpm, 4 °C) and the cleared supernatant was used for further purification steps.

Elution fractions containing the designed proteins were pooled and 333 µg of $His_6$-tagged SenP2 protease was added to each sample. The samples were transferred into a 3.5 kDa MWCO Spectra/Por3 Dialysis Membrane and dialysed overnight at 4 °C against 1 L dialysis buffer (same as wash buffer). The next day, all samples were removed from the 3.5 kDa MWCO Spectra/Por3 Dialysis Membrane and loaded again onto a 1 mL Protino Ni-NTA column (Macherey Nagel) at ca. 1 mL/min and 4 °C. The flow

through, which should contain the untagged designed proteins was collected. The $His_6$-tagged SenP2 protease, the $His_6$-Sumo3 fusion tag and any uncleaved fusion proteins should remain bound to the Ni-NTA beads. The flow through fractions were concentrated to 0.5 mL in an Amicon 3 kDa MWCO centrifugal filter unit (Merck Millipore). No protein was present in the flow through of the Amicon concentrators.

The concentrated samples obtained after the reverse Ni-NTA purification step were loaded onto a Superdex 75 10/300 size exclusion chromatography column (Cytiva) pre-equilibrated with SEC buffer (50 mM Na-phosphate pH 7.0, 100 mM NaCl). 0.5 mL elution fractions were collected and analysed via SDS-PAGE. The cleanest fractions for all 6 proteins were pooled, aliquoted, flash-frozen in liquid nitrogen and stored at −80 °C.

## Circular dichroism spectroscopy

Circular dichroism (CD) spectra were recorded to analyse the secondary structure of the proteins. CD spectrum of the sample (protein at 1 mg/mL) was recorded using a Jasco J-815 CD spectrophotometer using a 0.2 mm pathlength CD Quartz cuvette, at 20 °C, between 260 nm to 190 nm, with 10 accumulations, standard sensitivity, a bandwidth of 1 nm and a data-pitch of 0.1 nm. A buffer spectrum was first recorded as baseline and subtracted from the sample spectrum. The spectra were analysed using the BeStSel server with single spectrum analysis (https://bestsel.elte.hu/index.php) (Micsonai et al, 2018), with the CDSSTR (Johnson, 1999) method with the protein reference data 7 (Sreerama and Woody, 2000) using the DichroWeb server (http://dichroweb.cryst.bbk.ac.uk/html/process.shtml) (Miles et al, 2022) and with the Multivariate SSE analysis of the Jasco Spectra Manager software. A CD spectrum was calculated from the AF structure prediction of the protein using the KCD software (https://kcd.cinvestav.mx) (Jacinto-Méndez et al, 2024) and the PDBMD2CD software (https://pdbmd2cd.cryst.bbk.ac.uk) (Mavridis and Janes, 2016).

## Size exclusion chromatography coupled to multi-angle light scattering

A size exclusion chromatography coupled to multi-angle light scattering (SEC-MALS) experiment was performed to assess aggregation and the oligomeric state of the proteins. 50 µL of sample (protein at 1 mg/mL) was injected onto a Superdex 200 Increase 5/150 GL gel-filtration column (GE Healthcare) on an Agilent 1260 Infinity II HPLC system (Agilent) in 50 mM NaPhosphate pH 7.0 and 100 mM NaCl buffer at room temperature with a flow rate of 0.3 mL/min. The column was coupled to a MALS system (MiniDAWN and Optilab, Wyatt Technology). Data were analysed using the Astra 8.2.0 software (Wyatt Technology). The generic protein dn/dc value of 0.1850 was applied.

## Nuclear magnetic resonance structure determination

All nuclear magnetic resonance (NMR) spectra were recorded on Bruker Avance III HD 700 MHz and 900 MHz NMR spectrometers, equipped with cryogenically-cooled triple-resonance probes and a Bruker Avance III 600 MHz spectrometer equipped with a room temperature triple-resonance probe at a temperature of 298 K.

For protein resonance assignment, standard double and triple resonance through-bond experiments were recorded (Sattler et al, 1999). $^{15}$N and $^{13}$C-edited NOESY spectra (mixing time 120 ms) were performed to obtain distance restraints. Apodization weighted sampling was used for acquisition of triple resonance spectra (Simon and Köstler, 2019), and non-uniform sampling was applied for $^{15}$N-edited NOESY spectra. NMR data were processed using NMPipe (Delaglio et al, 1995) or in-house routines and analysed with Cara (http://cara.nmr.ch) or NMRViewJ (Keller, 2005; Johnson and Blevins, 1994). Secondary structure prediction from secondary chemical shifts ($C_\alpha$ and $C_\beta$) was done according to Wishart and Sykes (Wishart and Sykes, 1994) using random coil chemical shifts determined by Kjaergaard et al, 2011 (Kjaergaard et al, 2011).

For structure calculation using torsion angle dynamics with CYANA 3.98.15 (Güntert, 2009) dihedral angle restraints were derived from backbone chemical shifts using TALOS+ (Shen et al, 2009), and distance restraints were obtained from $^{15}$N- and $^{13}$C-edited NOESY spectra. NOEs were initially manually and later automatically assigned using the noe_assign function of CYANA 3.98.15.

## Validation of protein design using all-atom molecular dynamics simulations

Designed binder complexes were further rigorously validated by performing all-atom molecular dynamics (MD) simulations in explicit solvent and analysing the corresponding properties of structural flexibility and internal/intramonomeric and interfacial contacts. Furthermore, binding affinities were computed by combining an established structure-based predictor (Xue et al, 2016) with the conformational ensemble generated from the corresponding MD simulations.

The standard AMBER force field (ff14sb) was used to describe all protein parameters (Maier et al, 2015). Each protein or protein complex was solvated using atomistic TIP3P water (Jorgensen et al, 1983) with a minimum of 10 Å of padding to form a cubic periodic box and then electrically neutralized with an ionic concentration of 0.15 M NaCl that used standard ionic parameters (Joung and Cheatham, 2008).

A standardised minimisation, equilibration and simulation protocol consisting of several stages was developed for all systems. A set of restraints (RS) were applied to each system at specified stages of equilibration. These consisted of restraining all heavy (non-hydrogen) atoms of the proteins. Each system was subsequently minimised across four stages with 1500 steps (500 steepest-descent + 1000 conjugate-gradient) of minimization applying restraints RS with different force constants in each sequential stage: Stage 1: 10 kcal/mol Å$^2$, Stage 2: 5 kcal/mol Å$^2$, Stage 3: 1 kcal/mol Å$^2$, Stage 4: unrestrained. MD simulations were performed in all subsequent stages. The SHAKE algorithm was employed on all atoms covalently bonded to a hydrogen atom. A time-step of 2 fs was used. The long-range Coulomb interaction was handled using a GPU implementation of the particle mesh Ewald summation method (PME) (Essmann et al, 1995). A nonbonded cutoff distance of 10 Å$^2$ was used. In Stage 5, each system was heated from 10 K to 300 K in 1 ns and with RS ($k = 10$ kcal/mol Å$^2$). The temperature was subsequently maintained at 300 K using a Langevin thermostat with a damping constant of $\gamma = 5.0$ ps$^{-1}$ and in Stage 6, the systems equilibrated for 1 ns at constant volume, thus in the NVT ensemble.

Subsequently the pressure was maintained at 1 atm using a Berendsen barostat with a pressure relaxation time of $\tau_p = 1.0$ ps and the systems simulated in the NPT ensemble for 100 ps for each of the subsequent stages with RS: Stage 7: $k = 10$ kcal/mol Å$^2$, Stage 8: $k = 5$ kcal/mol Å$^2$, Stage 9: $k = 1$ kcal/mol Å$^2$, Stage 10: $k = 0.5$ kcal/mol Å$^2$. Finally, in Stage 11, RS restraints were removed and the systems simulated in the NPT ensemble for a further 5 ns. Following this, an ensemble of production simulations of $50 \times 4$ ns replicas was performed for each system in the NPT ensemble with the same conditions as Stage 11. Coordinate snapshots from production simulations were generated every 10 ps, resulting in an ensemble of trajectories consisting of 20,000 snapshots in aggregate per system.

Structural stability and flexibility were analysed by computing the root-mean squared deviation (RMSD) of the backbone atoms of the protein with respect to the initial predicted structure (after aligning the protein backbone) as well as the root-mean-square fluctuation (RMSF) with respect to the average structure in each production replica. In the case of complexes, this was carried out both on the overall complex and for individual monomers separately. The number of sidechain-sidechain intramonomer contacts were computed for each snapshot of the production MD, based on a heavy atom distance threshold of 4 Å. Similarly, interfacial contacts were computed for each interface in simulated protein complexes based on the same criteria.

Binding affinities of dimeric complexes were computed using the PRODIGY tool (Xue et al, 2016), an established structure-based affinity predictor that includes free energy contributions from interface contacts as well as the non-interacting surface. The dissociation constant ($K_D$) was computed for all 20,000 snapshots of the production MD ensemble and then averaged. A default threshold of 5.5 Å was used for the interface contact threshold between heavy-atoms of amino acids.

Molecular dynamics input and output files are deposited on Zenodo (https://doi.org/10.5281/zenodo.15208893). Input files are stored in "MD_inputs". Computed properties extracted from analysis of molecular dynamics simulations can be found in "MD_outputs".

# Data availability

The datasets and computer code produced in this study are available in the following databases: NMR data for RcaT-Sen2 inhibitor designs: BMRB (100aa_3_1: 34973; cpx-50-nr2-run_5_0: 52768) (https://bmrb.io/data_library/summary/index.php?bmrbId=34973, https://bmrb.io/data_library/summary/index.php?bmrbId=52768). NMR ensemble: PDB (100aa_3_1: 9HSA) (https://www.rcsb.org/structure/9HSA). All other data and code in this study: Zenodo (https://doi.org/10.5281/zenodo.15208893).

The source data of this paper are collected in the following database record: biostudies:S-SCDT-10_1038-S44320-025-00119-z.

# Peer review information

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

## Acknowledgements

JOK acknowledges support by the Bundesministerium für Bildung und Forschung (de.NBI project: 031A537B) and the Volkswagen Foundation (contract 95826). SKS acknowledges support by the Volkswagen Foundation "Experiment! Funding Initiative" (grant no. 93874-1). AT acknowledges European Molecular Biology Laboratory core funding. CGPV acknowledges support by HORIZON EUROPE Marie Skłodowska-Curie Actions COFUND (grant no. 847543). We thank EMBL Heidelberg for generously providing access to their HPC infrastructure (Laboratory et al, 2024) and the EMBL Protein Expression and Purification Core Facility (PEPCF) for their support.

## Author contributions

**Michael A Jendrusch**: Conceptualization; Software; Formal analysis; Investigation; Visualization; Methodology; Writing—original draft; Writing—review and editing. **Alessio L J Yang**: Conceptualization; Formal analysis; Validation; Investigation; Methodology; Writing—review and editing. **Elisabetta Cacace**: Formal analysis; Investigation; Writing—review and editing. **Jacob Bobonis**: Conceptualization; Investigation; Methodology; Writing—review and editing. **Carlos G P Voogdt**: Investigation; Writing—review and editing. **Sarah Kaspar**: Formal analysis; Validation. **Kristian Schweimer**: Formal analysis;

Investigation. **Cecilia Perez-Borrajero**: Formal analysis; Investigation; Writing—review and editing. **Karine Lapouge**: Formal analysis; Investigation; Writing—review and editing. **Jacob Scheurich**: Investigation. **Kim Remans**: Resources; Supervision; Methodology; Writing—review and editing. **Janosch Hennig**: Resources; Formal analysis; Supervision; Investigation; Methodology; Writing—review and editing. **Athanasios Typas**: Conceptualization; Resources; Supervision; Writing—review and editing. **Jan O Korbel**: Conceptualization; Resources; Supervision; Funding acquisition; Writing—original draft; Writing—review and editing. **S Kashif Sadiq**: Conceptualization; Resources; Formal analysis; Supervision; Funding acquisition; Investigation; Visualization; Methodology; Writing—original draft; Writing—review and editing.

Source data underlying figure panels in this paper may have individual authorship assigned. Where available, figure panel/source data authorship is listed in the following database record: biostudies:S-SCDT-10_1038-S44320-025-00119-z.

## Funding

## Disclosure and competing interests statement
SKS is the Founder and Chief Executive Officer of DenovAI Biotech Ltd. in which MJ, JOK and SKS have financial interests. MJ, JOK, and SKS are co-inventors on US Provisional Patent Application No. 63/329,522/PCT patent application (published as WO 2023/198726 A1 on 19.10.2023). JOK and AT are members of the Advisory Editorial Board of Molecular Systems Biology. This has no bearing on the editorial consideration of this article for publication.

# Expanded View Figures

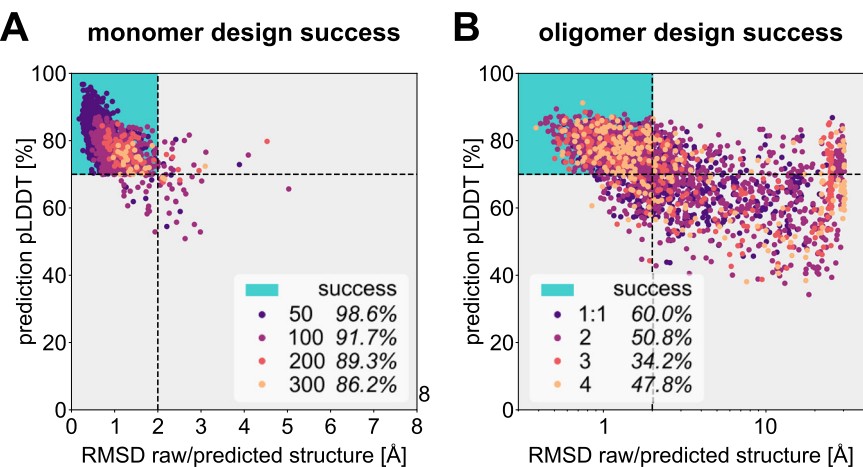

**Figure EV1.  ESMfold de novo design success.**

(A, B) Scatter plots of redesigned sequence scRMSD/pLDDT using ESMfold for monomers (A) and oligomers (B). Designs are coloured by number of amino acids and percentages of successful designs are reported. The turquoise region and dotted lines indicate the region of success in terms of scRMSD/pLDDT (scRMSD < 2.0 Å, pLDDT > 70.0).

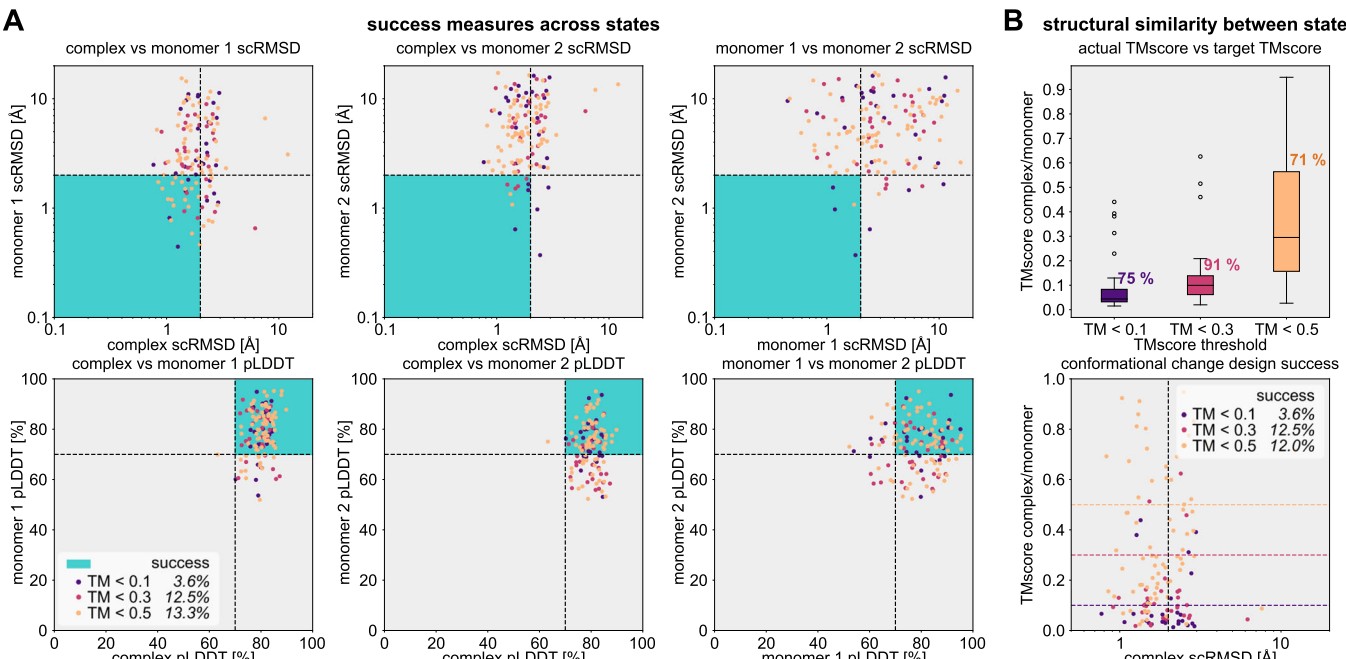

**Figure EV2. Conformational change design success.**

(A) Scatter plots of scRMSD (top) and pLDDT (bottom) using AlphaFold, comparing the complex state and both monomers of a conformational change design. Designs are coloured by their TM-score threshold (0.1, $N = 28$; 0.3, $N = 32$; or 0.5, $N = 75$) and the region of successful designs in terms of scRMSD or pLDDT is marked in blue. For each TM-score threshold, the percentage of successful designs with complex scRMSD < 2.0 Å, pLDDT > 70 and monomer scRMSD < 3.0 Å are reported. (B) Distribution of TM-scores between the complex and monomeric state predicted using AlphaFold for each TM-score threshold. (top) as a box-plot reporting the percentage of designs with TM-score below threshold. (bottom) as a scatter plot comparing scRMSD and TM-score. Designs are coloured by TM-score threshold and correspondingly coloured dotted lines mark the position of each threshold in the scatter plot (0.1, $N = 28$; 0.3, $N = 32$; 0.5, $N = 75$). The centre line of each box in the box plot corresponds to the median, box edges to the upper and lower quartiles, and whiskers to the lowest and highest data points excluding outliers (within 1.5 times the interquartile range).

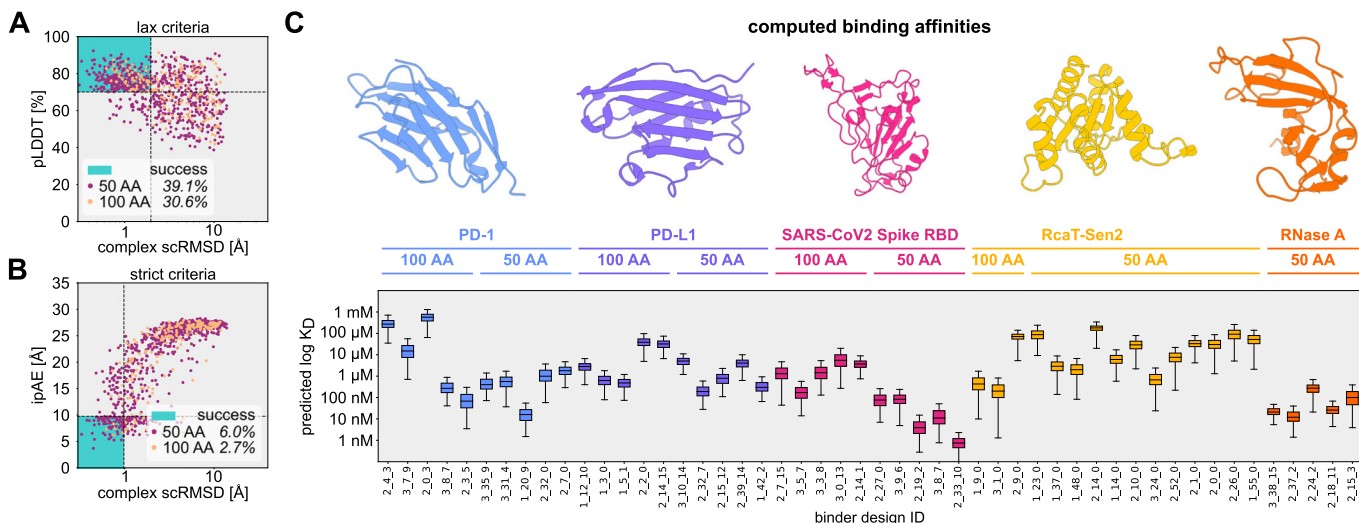

**Figure EV3. Binder design success rate.**

(A, B) Scatter plots of redesigned sequence scRMSD, pLDDT and interface pAE for binder designs to all target proteins considered in this work. Designs are coloured by number of amino acids (50 AA: $N = 715$, 100 AA: $N = 203$) and percentages of successful designs are reported. A shows lax success criteria, B shows strict success criteria. (C) Box plot of predicted dissociation constants for a set of selected binder designs over $N = 20{,}000$ molecular dynamics snapshots. Designs are grouped by target protein and the structures of each target are displayed above. The centre line of each box corresponds to the median, box edges to the upper and lower quartiles, and whiskers to the lowest and highest data points excluding outliers (within 1.5 times the interquartile range).

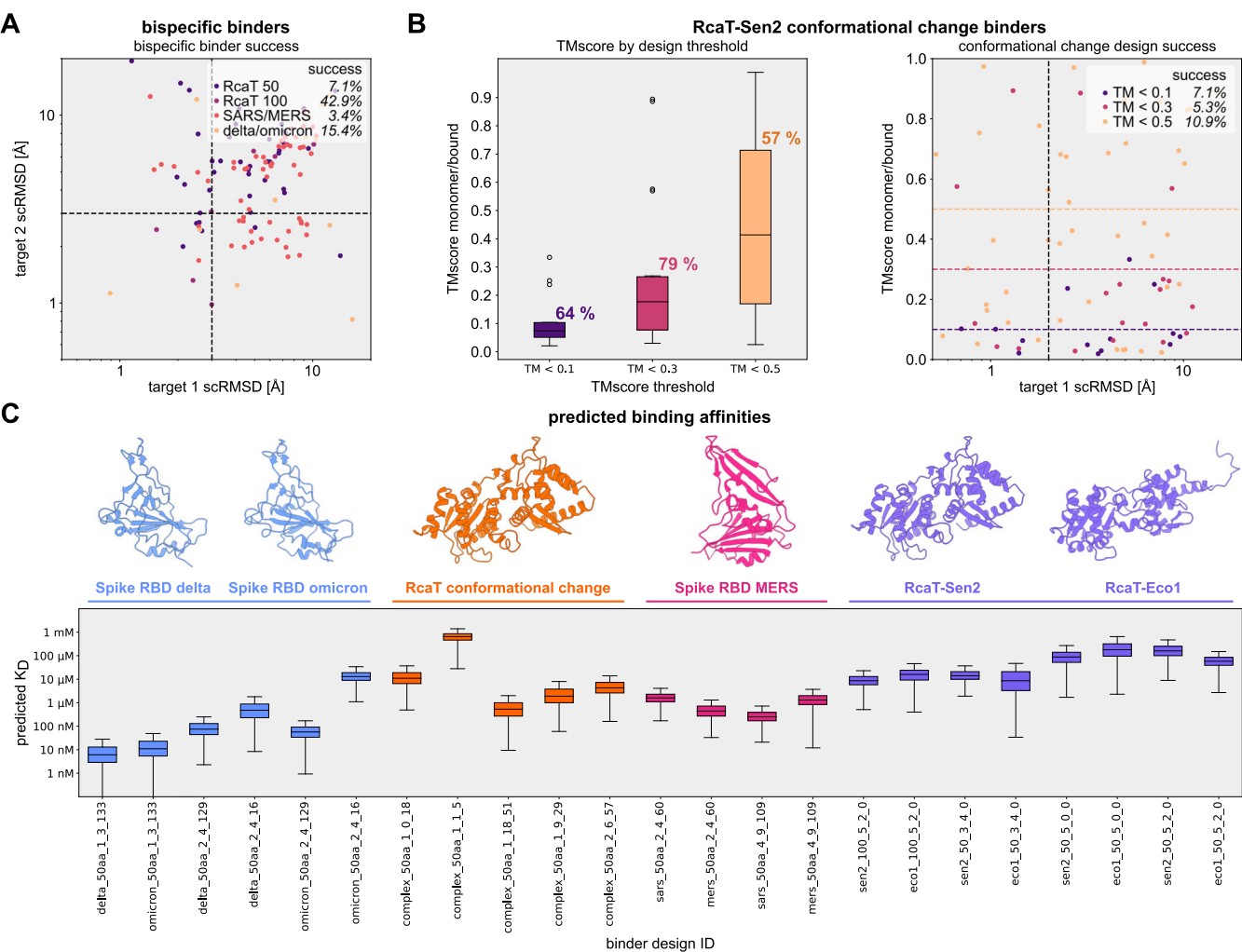

**Figure EV4. Multistate binder design.**

(A) Scatter plot of complex scRMSD for two targets of a bispecific binder design. Designs are coloured by target and number of amino acids (RcaT-Sen2/Eco1 50 AA: $N = 42$, 100 AA: $N = 7$; SARS/MERS Spike RBD 50 AA: $N = 58$; delta/omicron Spike RBD 50 AA: $N = 13$). Dotted lines represent a relaxed threshold for bispecific binder design success of 3.0 Å. Percentage of successful designs with pLDDT > 70 and scRMSD < 3.0 Å are reported for each pair of targets and designed binder size. (B) Statistics of proteins designed to change conformation upon RcaT-Sen2 binding. (left) Scatter plot of complex scRMSD of the bound state (x-axis) compared to the binder TM-score between the bound and unbound state. Designs are coloured by the maximum TM-score threshold used in their fitness function (TM < 0.1: $N = 14$, TM < 0.3: $N = 19$, TM < 0.5: $N = 46$). Coloured dotted lines represent these respective TM-score thresholds. The black dotted line represents the threshold for design success of scRMSD < 2.0 Å. Percentage of successful designs per TM-score threshold is reported. (right) Box plot of TM-scores for structures predicted using AF for each TM-score threshold used in the design fitness function. Percentage of designs below the threshold is reported. The centre line of each box corresponds to the median, box edges to the upper and lower quartiles, and whiskers to the lowest and highest data points excluding outliers (within 1.5 times the interquartile range). (C) Box plot of predicted dissociation constants for a set of selected multi-state binder designs ($N = 20,000$ molecular dynamics snapshots per box). Designs are grouped by target protein and the structures of each target are displayed above. Boxes, centre line and whiskers as in (B).

