## [Peer Review File · Molecular Systems Biology]

AlphaDesign: A de novo protein design framework based on AlphaFold

Michael Jendrusch, Alessio Yang, Elisabetta Cacace, Jacob Bobonis, Carlos Voogdt, Sarah Kaspar, Kristian Schweimer, Cecilia Perez-Borrajero, Karine Lapouge, Jacob Scheurich, Kim Remans, Janosch Hennig, Athanasios Typas, Jan Korbelt, and S. Kashif Sadiq

Corresponding author(s): Jan Korbelt (jan.korbelt@embl.de) , S. Kashif Sadiq (kashif@denovai.com)

Review Timeline:

Submission Date:	8th Apr 25
Editorial Decision:	8th Apr 25
Revision Received:	23rd Apr 25
Editorial Decision:	2nd May 25
Revision Received:	5th May 25
Accepted:	6th May 25

Editor: Jingyi Hou

Transaction Report:

This manuscript was transferred to Molecular Systems Biology following peer review at another journal.

8th Apr 2025

Manuscript Number: MSB-2025-13029P

Title: AlphaDesign: A de novo protein design framework based on AlphaFold

Author: Jan Korbel

Dear Jan,

Thank you for submitting your presubmission inquiry. I have now had the opportunity to review the revised manuscript along with your point-by-point response to the reviewers' comments from the other journal. Overall, we think the study makes a significant and unique contribution to the field, and we will be able to accept it for publication pending a minor revision, as outlined below.

Based on the provided point-by-point response, we find that the technical issues and related concerns have been adequately addressed. Regarding Reviewer #4's comments on the overall novelty of the study, it appears that you have already provided appropriate justification and clarification, and have made the necessary revisions in response to their feedback. As for the remaining comment about high-throughput experimental validation, while we do not consider this mandatory for publication in Molecular Systems Biology, we recommend acknowledging it as a limitation and suggesting it as a potential direction for future research.

On a more editorial level, please address the following issues :

- Please provide a .docx formatted version of the manuscript text (including legends for main figures, EV figures and tables). Please make sure that the changes are highlighted to be clearly visible.
- Please provide individual production quality figure files as .eps, .tif, .jpg (one file per figure).
- Please provide a .docx formatted letter INCLUDING the reviewers' reports and your detailed point-by-point responses to their comments. As part of the EMBO Press transparent editorial process, the point-by-point response is part of the Review Process File (RPF), which will be published alongside your paper.
- Please note that all corresponding authors are required to supply an ORCID ID for their name upon submission of a revised manuscript.
- We replaced Supplementary Information with Expanded View (EV) Figures and Tables that are collapsible/expandable online (see examples in <http://msb.embopress.org/content/11/6/812>). A maximum of 5 EV Figures can be typeset. EV Figures should be cited as 'Figure EV1, Figure EV2' etc... in the text and their respective legends should be included in the main text after the legends of regular figures. Additional Tables/Datasets should be labeled and referred to as Table EV1, Dataset EV1, etc. Legends have to be provided in a separate tab in case of .xls files. Alternatively, the legend can be supplied as a separate text file (README) and zipped together with the Table/Dataset file. For the figures and tables that you do NOT wish to display as Expanded View figures, they should be bundled together with their legends in a single PDF file called *Appendix*, which should start with a short Table of Content. Each legend should be below the corresponding Figure/Table in the Appendix. Appendix figures and tables should be referred to in the main text as: "Appendix Figure S1, Appendix Figure S2, Appendix Table S1" etc. See detailed instructions regarding expanded view here: <https://www.embopress.org/page/journal/17444292/authorguide#expandedview>.
- Before submitting your revision, primary datasets (and computer code, where appropriate) produced in this study need to be deposited in an appropriate public database (see <http://msb.embopress.org/authorguide-dataavailability> <https://www.embopress.org/page/journal/17444292/authorguide#dataavailability>). Please remember to provide a reviewer password if the datasets are not yet public. The accession numbers and database should be listed in a formal "Data Availability" section (placed after Materials & Method) that follows the model below (see also <https://www.embopress.org/page/journal/17444292/authorguide#dataavailability>). Please note that the Data Availability Section is restricted to new primary data that are part of this study.
Data availability
The datasets (and computer code) produced in this study are available in the following databases:
 - RNA-Seq data: Gene Expression Omnibus GSE46843 (<https://www.ncbi.nlm.nih.gov/geo/query/acc.cgi?acc=GSE46843>)
 - [data type]: [name of the resource] [accession number/identifier/doi] ([URL or identifiers.org/DATABASE:ACCESSION])*** Note - All links should resolve to a page where the data can be accessed. ***
- At EMBO Press we ask authors to provide source data for the main figures. Our source data coordinator will contact you to

discuss which figure panels we would need source data for and will also provide you with helpful tips on how to upload and organize the files.

- Our journal encourages inclusion of *data citations in the reference list* to directly cite datasets that were re-used and obtained from public databases. Data citations in the article text are distinct from normal bibliographical citations and should directly link to the database records from which the data can be accessed. In the main text, data citations are formatted as follows: "Data ref: Smith et al, 2001". In the Reference list, data citations must be labeled with "[DATASET]". A data reference must provide the database name, accession number/identifiers and a resolvable link to the landing page from which the data can be accessed at the end of the reference. Further instructions are available at .

- We updated our journal's competing interests policy in January 2022 and request authors to consider both actual and perceived competing interests. Please review the policy <https://www.embopress.org/competing-interests> and update your competing interests if necessary. Please use the heading "Disclosure statement and competing interests". Please also add "Jan Korbel is a member of the Advisory Editorial Board of Molecular Systems Biology. This has no bearing on the editorial consideration of this article for publication."

- All Materials and Methods need to be described in the main text using our 'Structured Methods' format. According to this format, the Methods section includes a Reagents and Tools Table (listing key reagents, experimental models, software and relevant equipment and including their sources and relevant identifiers) followed by a Methods and Protocols section describing the methods, ideally using a step-by-step protocol format. The aim is to facilitate adoption of the methodologies across labs.

Please download and fill our Reagents and Tools Table template (.docx), which you can find in our author guidelines: <https://www.embopress.org/page/journal/17444292/authorguide#structuredmethods>.

An example of a Method paper with Structured Methods can be found here: <https://www.embopress.org/doi/10.15252/msb.20178071>.

-Regarding data quantification:

Please ensure to specify the name of the statistical test used to generate error bars and P values, the number (n) of independent experiments (please specify technical or biological replicates) underlying each data point and the test used to calculate p-values in each figure legend. Discussion of statistical methodology can be reported in the materials and methods section, but figure legends should contain a basic description of n, P and the test applied.

Graphs must include a description of the bars and the error bars (s.d., s.e.m.).

- Please provide a "standfirst text" summarizing the study in one or two sentences (approximately 250 characters, including space), three to four "bullet points" highlighting the main findings and a "synopsis image" (550px width and 400-600 px height, PNG format) to highlight the paper on our homepage.

Here are a couple of examples:

<https://www.embopress.org/doi/10.15252/msb.20199356>

<https://www.embopress.org/doi/10.15252/msb.20209475>

<https://www.embopress.org/doi/10.15252/msb.209495>

When you resubmit your manuscript, please download our CHECKLIST (<https://www.embopress.org/pb-assets/embo-site/EMBO%20Press%20Author%20Checklist-1642513524327.xlsx>) and include the completed form in your submission.

Please note that the Author Checklist will be published alongside the paper as part of the transparent process (<https://www.embopress.org/page/journal/17444292/authorguide#transparentprocess>).

Click on the link below to submit your revised paper.

Thank you for submitting this interesting paper to Molecular Systems Biology.

Kind regards,
Jingyi

If you do choose to resubmit, please click on the link below to submit the revision online before 8th May 2025.

IMPORTANT: When you send your revision, we will require the following items:

1. the manuscript text in LaTeX, RTF or MS Word format
2. a letter with a detailed description of the changes made in response to the referees. Please specify clearly the exact places in the text (pages and paragraphs) where each change has been made in response to each specific comment given
3. three to four 'bullet points' highlighting the main findings of your study
4. a short 'blurb' text summarizing in two sentences the study (max. 250 characters)
5. a 'thumbnail image' (550px width and max 400px height, Illustrator, PowerPoint or jpeg format), which can be used as 'visual title' for the synopsis section of your paper.
6. Please include an author contributions statement after the Acknowledgements section (see <https://www.embopress.org/page/journal/17444292/authorguide#manuscriptpreparation>)
7. Please complete the CHECKLIST available at (<https://bit.ly/EMBOPressAuthorChecklist>). Please note that the Author Checklist will be published alongside the paper as part of the transparent process (<https://www.embopress.org/page/journal/17444292/authorguide#transparentprocess>).
8. When assembling figures, please refer to our figure preparation guideline in order to ensure proper formatting and readability in print as well as on screen:
<https://bit.ly/EMBOPressFigurePreparationGuideline>
See also figure legend guidelines: <https://www.embopress.org/page/journal/17444292/authorguide#figureformat>
9. Please note that corresponding authors are required to supply an ORCID ID for their name upon submission of a revised manuscript (EMBO Press signed a joint statement to encourage ORCID adoption). (<https://www.embopress.org/page/journal/17444292/authorguide#editorialprocess>)
Currently, our records indicate that there is no ORCID associated with your account.

Please click the link below to provide an ORCID:

Link Not Available

10. Include a Reagents and Tools Table as part of the Methods section, which can be downloaded from our author guidelines (<https://www.embopress.org/page/journal/17444292/authorguide#structuredmethods>)

*** PLEASE NOTE *** As part of the EMBO Press transparent editorial process initiative (see our Editorial at <https://dx.doi.org/10.1038/msb.2010.72> , Molecular Systems Biology will publish online a Review Process File to accompany accepted manuscripts. When preparing your letter of response, please be aware that in the event of acceptance, your cover letter/point-by-point document will be included as part of this File, which will be available to the scientific community. More information about this initiative is available in our Instructions to Authors. If you have any questions about this initiative, please contact the editorial office (msb@embo.org).

The authors addressed the minor editorial issues.

2nd May 2025

Manuscript Number: MSB-2025-13029

Title: AlphaDesign: A de novo protein design framework based on AlphaFold

Author: Michael Jendrusch

Alessio Yang

Elisabetta Cacace

Jacob Bobonis

Carlos Voogdt

Sarah Kaspar

Kristian Schweimer

Cecilia Perez-Borrajero

Karine Lapouge

Jacob Scheurich

Kim Remans

Janosch Hennig

Athanasios Typas

Jan Korbelt

S. Kashif Sadiq

Dear Jan,

Thank you for sending us your revised manuscript. We are now satisfied with the performed revisions. Before we can proceed with formal acceptance, we kindly ask you to address the remaining minor issues listed below:

1. Please remove the "Authors' Contribution" section from the manuscript.
2. Provide up to five keywords in the manuscript.
3. Ensure that both corresponding authors are labeled on the title page, along with their email addresses.
4. Rename the "Disclosure statement and competing interests" to "DISCLOSURE AND COMPETING INTERESTS STATEMENT".
5. Please ensure that all datasets will be made publicly available upon the acceptance of the manuscript.
6. Figure callouts:
 - all callouts should appear in sequential order.
 - There is a callout for Supp. Table 3, which does not appear exist. Please review and correct this discrepancy.
 - Similarly, there is a callout for Figure 2E, but only 2A-C are present. This inconsistency needs to be resolved as well.
7. Please address the following issues related to figure legends:
 - Please note that the box plots need to be defined in terms of minima, maxima, centre, bounds of box and whiskers, and percentile in the legends of figures 4C, D; 7D, EV2 B, EV3 C, EV4 B, C; S1A, B; S2A, B.
 - Please note that information related to n is missing in the legends of figures 4C, D; S1A, B; S6
 - Please note that the error bars are not defined in the legend of figure S6.
8. Please revise the manuscript to follow the correct section order: Title page - Abstract & Keywords - Introduction - Results - Discussion - Methods - Data Availability - Acknowledgements - Disclosure and Competing Interests Statement - References - Figure Legends - Table(s) - Expanded View Figure Legends.

When you resubmit your manuscript, please download our CHECKLIST (<https://bit.ly/EMBOPressAuthorChecklist>) and include the completed form in your submission. *Please note* that the Author Checklist will be published alongside the paper as part of the transparent process (<https://www.embopress.org/page/journal/17444292/authorguide#transparentprocess>)

Click on the link below to submit your revised paper.

Kind regards,
Jingyi

Jingyi Hou, PhD
Senior Editor
Molecular Systems Biology

*** PLEASE NOTE *** As part of the EMBO Press transparent editorial process initiative (see our Editorial at <https://dx.doi.org/10.1038/msb.2010.72> , Molecular Systems Biology will publish online a Review Process File to accompany accepted manuscripts. When preparing your letter of response, please be aware that in the event of acceptance, your cover letter/point-by-point document will be included as part of this File, which will be available to the scientific community. More information about this initiative is available in our Instructions to Authors. If you have any questions about this initiative, please contact the editorial office (msb@embo.org).

All editorial and formatting issues were resolved by the authors.

6th May 2025

Manuscript number: MSB-2025-13029R

Title: AlphaDesign: A de novo protein design framework based on AlphaFold

Dear Jan,

Thank you again for sending us your revised manuscript. We are now satisfied with the modifications made and I am pleased to inform you that your paper has been accepted for publication.

If you have any questions, please do not hesitate to contact the Editorial Office. Thank you for your nice contribution to Molecular Systems Biology.

Kind regards,
Jingyi

Jingyi Hou, PhD
Senior Editor
Molecular Systems Biology
